# Towards a Comprehensive Definition of Pandemics and Strategies for Prevention: A Historical Review and Future Perspectives

**DOI:** 10.3390/microorganisms12091802

**Published:** 2024-08-30

**Authors:** Ricardo Augusto Dias

**Affiliations:** School of Veterinary Medicine, University of Sao Paulo, Av. Prof. Dr. Orlando Marques de Paiva, 87, São Paulo 05508-270, Brazil; ricardodias@usp.br

**Keywords:** pandemics, smallpox, plague, cholera, influenza, AIDS, coronavirus

## Abstract

The lack of a universally accepted definition of a pandemic hinders a comprehensive understanding of and effective response to these global health crises. Current definitions often lack quantitative criteria, rendering them vague and limiting their utility. Here, we propose a refined definition that considers the likelihood of susceptible individuals contracting an infectious disease that culminates in widespread global transmission, increased morbidity and mortality, and profound societal, economic, and political consequences. Applying this definition retrospectively, we identify 22 pandemics that occurred between 165 and 2024 AD and were caused by a variety of diseases, including smallpox (Antonine and American), plague (Justinian, Black Death, and Third Plague), cholera (seven pandemics), influenza (two Russian, Spanish, Asian, Hong Kong, and swine), AIDS, and coronaviruses (SARS, MERS, and COVID-19). This work presents a comprehensive analysis of past pandemics caused by both emerging and re-emerging pathogens, along with their epidemiological characteristics, societal impact, and evolution of public health responses. We also highlight the need for proactive measures to reduce the risk of future pandemics. These strategies include prioritizing surveillance of emerging zoonotic pathogens, conserving biodiversity to counter wildlife trafficking, and minimizing the potential for zoonotic spillover events. In addition, interventions such as promoting alternative protein sources, enforcing the closure of live animal markets in biodiversity-rich regions, and fostering global collaboration among diverse stakeholders are critical to preventing future pandemics. Crucially, improving wildlife surveillance systems will require the concerted efforts of local, national and international entities, including laboratories, field researchers, wildlife conservationists, government agencies and other stakeholders. By fostering collaborative networks and establishing robust biorepositories, we can strengthen our collective capacity to detect, monitor, and mitigate the emergence and transmission of zoonotic pathogens.

## 1. Introduction

There is no agreed definition of a pandemic in the scientific literature [1]. Classically, a pandemic is defined as “an epidemic that occurs worldwide, or over a very large area, crosses international borders, and usually affects a large number of people” [2,3]. However, this definition ignores population immunity, pathogen virulence, the severity of symptoms [4], and even the metapopulation structure [5]. Until the COVID-19 pandemic, the World Health Organization (WHO) defined a pandemic simply as a “new disease with worldwide spread” [6]. This term has not been updated by the WHO since. On the other hand, Madhav et al. [7] define a pandemic as “large-scale outbreak of infectious disease that greatly increases morbidity and mortality over a large geographic area and causes significant economic, social, and political disruption”. The latter definition adds to the classical definition the need to prioritize the consequences of pandemics, be they economic, social, or political.

These definitions are at best qualitative and vague, using terms such as “very large area”, “a large number of people”, “large-scale” or “greatly increase”. Some available definitions of a pandemic at best approach the quantitative aspect by using terms such as “sustained transmission” or “emergence of new pathogens” [1]. In any case, the likelihood of pandemics has increased in recent centuries due to global transportation and integration, urbanization, changes in land use, exploitation of the natural environment, and climate change. These trends have simultaneously increased the need for international preparedness for new pandemics and the capacity of health systems [7]. For this reason, in the event of an outbreak, it would be important to estimate the likelihood of it becoming a pandemic, in order to predict its impact on health services and the global economy.

In addition to the speed of spread, a pandemic concept should take into account the likelihood that susceptible individuals will become infected. In this sense, some vector-borne or seasonal diseases cannot be transmitted to all susceptible individuals in a given period of time because not all regions of the world would be under the same risk of transmission, either because of the absence of vectors or because of favorable climatic conditions. Therefore, if the vectors are not cosmopolitan, i.e., regionally specific, or if the disease requires a specific climatic condition as a determining factor, the disease in question should not be considered a pandemic. For example, the vectors of dengue fever, chikungunya and zika, the mosquitoes of the *Aedes* genus, mainly *Aedes aegypti* and *A. albopictus*, are currently distributed in tropical and subtropical areas. Although they have a tendency to expand their range in the Northern Hemisphere in the coming years due to environmental changes, they do not occur in temperate and cold climates [8] and therefore could not transmit these diseases in these regions. Plague, on the other hand, can be considered a pandemic because both the vector (the flea *Xenopsylla cheopis*) [9] and its reservoirs (commensal rats) [10] are cosmopolitan.

Based on the above, a tentative definition of a pandemic should consider the pandemic behavior of the disease, when there is a global spread, i.e., after affecting a specific region, the outbreak spreads locally and then to different continents with sustained human-to-human transmission, leading to an increased morbidity and/or mortality, and causing social, economic, and/or political disruption. Finally, a pandemic can be caused by either emerging or re-emerging pathogens.

By this tentative definition, there have been 22 documented pandemics in human history, two of which caused by smallpox (Antonine Plague and American), three by plague (Justinian Plague, Black Death and the Third Plague), seven by cholera, six by influenza (two Russian Flu, Spanish Flu, Asian Flu, Hong Kong Flu and Swine Flu), AIDS and three by coronaviruses (SARS, MERS, and COVID-19). It is important to remember that the determinants of these diseases have changed over the years, but still the pandemics have found the right time to occur. In addition, the consequences of pandemics have depended on the historical context of when they occurred, but although been less lethal in the last century, they have generally been significant, with disruptive effects on the economy, society and politics.

Lack of information in the literature can make it difficult to classify even a significant health event as a pandemic, for example, because it is difficult to define the pathogen involved. An example is the Athenian Plague, described by Thucydides, which broke out in Ethiopia between 430 and 26 BC. The pathogen has not been identified, nor has the death toll [11], but it is thought to have been caused by the influenza virus [12]. Another example is a global flu spread that originated in Asia and traveled from southeast to northwest to Europe between 1557 and 1558, presumably caused by the influenza virus. Recurrence with peak mortality between 1558 and 1561, occurred first in large urban centers, then in smaller and smaller cities, and finally in the interior. By 1580, it had reached major urban centers around the world [12].

Some scientific papers have compiled the pandemics that have occurred in history [13], without mentioning inclusion criteria, most of which were published during the COVID-19 pandemic [11,14]. This is the first work to consider inclusion criteria in a pandemic compilation and to provide insights into changes in epidemiological patterns that may be useful in preparing for new pandemics.

## 2. Smallpox

Smallpox has struck humanity in devastating waves for most of our modern history. It has caused more suffering and death than any other disease. The disease may have originated in East Africa between 3000 and 4000 years ago [15,16], from where it may have been introduced to India by ancient Egyptian traders [17]. The first evidence of lesions compatible with smallpox was found in Egyptian mummies from 1570 to 1085 BC, including the Pharaoh Ramses V, who died in 1156 BC [17,18]. The disease was also described in China in 1122 BC and in Sanskrit medical texts from India around 1500 BC [17,18].

Smallpox is an acute infectious disease caused by the variola virus (VARV), a DNA virus of the genus *Orthopoxvirus* and family Poxviridae that infects humans [15]. The genus *Orthopoxvirus* includes several other viruses that infect humans and animals, such as the monkeypox virus (MPXV), vaccinia virus (VACV), and the viruses most closely related to VARV, camelpox virus (CMLV) and taterapox virus (TATV) (which infects gerbils) [16]. Although the recent evolutionary history of VARV is known, its origin is still unknown [16], but it may have been zoonotic. The two lineages of VARV, P-I (“variola major”) and P-II (“variola minor”), diverged just before the end of the 17th century (when vaccination began, see below), but experienced severe bottlenecks, less severe for P-II, probably because this lineage was less virulent, maintained at endemic levels and in low transmission, and for these reasons, less recognizable and less susceptible to control measures (isolation and quarantine) [16].

The virus is airborne and can also be spread by fomites such as clothing and bedding [19]. Congenital infection also occurs in pregnant women, causing abortions or stillbirths [20]. Most smallpox outbreaks have been caused by “variola major” with a case fatality rate of up to 30% [19]. Occasionally, some outbreaks have resulted in case fatality rates as low as 1%, which were caused by “variola minor” [20]. The incubation period is between 10 and 14 days, usually 12 days. The early symptom of smallpox is fever, and if it develops, the infected person becomes contagious. Other early symptoms, in addition to fever, include headache, chills, and back pain, and less commonly, vomiting [20]. After two to four days, rashes more commonly appear on the face. After two days of rash all over the body, the macules raise to the surface of the skin due to fluid effusion and become vesicles. By day 7, the vesicles become pustules, and by week 2, the fluids are absorbed and the lesion dries, forming crusts that dry and fall off after approximately three weeks [19,20]. The fever persists until the crusts form. When the crusts fall off by the end of the 3rd week of illness, the person is no longer infectious, but may be left with blindness, sterility, and disfigurement. In fatal cases, death occurs between 10 and 16 days after the onset of the symptoms [20]. Immunity following the infection declines slowly over time and remains virtually stable for decades after recovery [21].

Smallpox has been linked to disruptions in the development of Western civilizations, for example the decline of the Roman Empire coincides with the first documented pandemics, the Antonine Plague.

### 2.1. Antonine Plague (165–180 AD)

This was the first pandemic ever recorded, but probably not the first that humans have faced in history. The smallpox pandemic began in the city of Seleucia (Mesopotamia) in late 165 AD during a campaign by the Roman army and spread to Rome the following year [22]. It lasted until 180 AD and became a disruptive event in Roman history [23]. It is estimated that between 7 and 10 million people died during the pandemic [23], approximately 20% of the population of the Roman Empire [24], or approximately 5% of the world’s population based on the global population estimate in the first year of the pandemic [25] (Figure 1). One of the victims was the Emperor Lucio Vero himself in 166 AD [26], and the plague was named after his co-ruler, Marcus Aurelius Antoninus, because of the end of the Pax Romana era.

The impact of this pandemic was so severe that it reduced military conscription, food production and economic activity. Conversely, traditions were affected, paving the way for the spread of monotheistic religions such as Mithraism and Christianity. This crisis also allowed the barbarian invasion of the Empire. In conclusion, the Antonine Plague paved the way for the decline of the Roman Empire and ultimately the fall of the West in the 5th century AD [27].

### 2.2. Smallpox Taking over the World

A well-documented history of smallpox comes from Japan. The disease was introduced by Korean merchants and Buddhist missionaries in 735 AD. The first smallpox epidemic killed approximately 30% of the Japanese population. A further 27 epidemics were recorded up to 1206 AD, mainly affecting children. The long-term effects of smallpox in Japan were a shortage of labor, a decline in agricultural production, and a reduction in tax revenue for the empire [28].

As the population grew, smallpox became more prevalent in cities worldwide. The demographics of the disease were the same, affecting mostly children and killing approximately 30% of those infected. The predominance of children as mostly affected suggests an endemic behavior of the VARV [20]. By the 16th century, smallpox was already a major cause of morbidity and mortality in Europe, Southeast Asia, China, and India. Europe was particularly important in the global spread of smallpox that accompanied the successive waves of exploration and colonization [20].

### 2.3. Smallpox in the Americas (1520–1880 AD)

Smallpox first appeared in the New World, in 1507 AD on the island of Hispaniola, introduced by the Spanish conquistadors. An epidemic ensued, wiping out tribes, but eventually exhausting itself. In 1517, an outbreak occurred among the African slaves and spread to the Amerindians, killing approximately one-third of the native population of Hispaniola. Smallpox managed to spread to Cuba in 1518 AD and to Puerto Rico in 1519 AD, killing half of the native population of those islands. In 1519 AD, Hernán Cortés sailed from Cuba on an expedition to explore and conquer what is now Mexico. In 1520 AD, the governor of Cuba sent Pánfilo de Narváez to replace Cortés, bringing with him an African slave who had contracted smallpox [20]. This was the beginning of the most devastating event for the Aztec Empire. From the coast, the disease reached the capital, Tenochtitlán, in five months. The loss of the chiefs Mexixcatl and Cuitláhuac and men, to smallpox, rather than the Spaniards (who were immune), was demoralizing and allowed for a quick Spanish conquest [20]. From Mexico, the disease spread to the Yucatan Peninsula, decimating the dense Mayan population [20]. The estimated Amerindian population before the Spanish occupation was 25 million [20], and one-fifth to one-third (5 to 8 million people) died of smallpox within 6 months [29].

The disease also spread to the Inca Empire even before the arrival of the Spanish conquistador Francisco Pizarro. Between 1524 and 1527 AD, the smallpox is believed to have killed 200,000 of the empire’s 6 million people, including the emperor Huayna Capac and his heir Ninan Cuyochi. A civil war ensued, making it easy for a small band of Spanish conquistadors to enter Cuzco, in 1533 AD. No part of South America was spared, and the disease even reached the interior of Brazil, killing more than 60,000 Indians [20].

In North America in the 17th century, the population was sparsely distributed, living much like hunter-gatherers or primitive farmers, estimated at approximately 3 million Native Americans [20]. The continent was colonized by the British, French, and Dutch. The first smallpox outbreak occurred on the Massachusetts coast between 1617 and 1619, killing most of the natives in the region and opening the way for the European settlers. The outbreaks became more frequent as the disease spread inland, affecting people of all ages, compatible with infection of a naïve population. Because the settlements were smaller than the cities of Central America, the outbreaks soon disappeared, and the reintroduction of smallpox would occur only after long intervals, with the accumulation of new susceptible individuals, and usually by ship arriving on the East coast. The distinct smallpox endemicity of the colonies and Great Britain is demonstrated by the fact that young colonists who had lived in smallpox-free areas and went to Great Britain for further study contracted the disease. The risk of infection in Britain was one of the reasons for funding colleges and universities in the colonies [20].

As military activity increased in the late 17th century, the British and French, using Indian allies, waged war in what is now Canada. Sometimes, epidemics were deliberately started. For example, in 1763, during the French and Indian War in the British colonies of North America, British soldiers distributed blankets that had been used by the sick to the native population in an act of biological warfare. Smallpox outbreaks declined until the end of the 1880s, just as railroads began to spread the disease across the continent [20].

As a result of the 360-year circulation of smallpox in the Americas, the Amerindian population of the Caribbean islands was replaced by African slaves, while the Aztec and Inca populations on the mainland survived to the present day and comprise approximately half of the current population. In contrast, in North and South America, the Native Americans make up only a small fraction of the population [20]. Several other diseases were introduced to the Americas by European conquerors. A hemorrhagic syndrome, called “cocoliztli” [30], probably caused by the bacterium *Salmonella enterica* paratyphi C [31], associated with typhus, measles, and/or smallpox, killed between 7 and 17 million Mesoamericans in two outbreaks, between 1545 and 1576 AD [30]. The resulting social and economic disruption is the likely cause of the abandonment of several cities and the decline of the Mesoamerican civilizations. By the end of the 19th century, smallpox and other diseases may have contributed to the decline of up to 90% of the indigenous population of the Americas [20,29,30], or 6.4% of the world’s population (using 1500 AD as reference) [25] (Figure 2).

### 2.4. Steps to the Smallpox Eradication

The observation that smallpox survivors were pockmarked and never became re-infected was the starting point for the first attempts at prevention. In addition, those who were accidentally infected by scratching their skin, developed a mild form of the disease. These observations led to the administration of pustular fluid or dried scabs to people who had never had smallpox, resulting in a much milder disease. This practice, known as “variolation”, began independently in China (by blowing ground dried scabs into the nostrils) and India (by cutaneous inoculation) in the 10th century. Cutaneous variolation was introduced in Egypt in the 13th century. Variolation was widely used in China in the 17th century, but it was not until the 18th century that these ancient Chinese methods were introduced to Europe and its colonies. Variolation produced satisfactory immunity, although it caused a mortality rate of 0.5–2% [20].

In the 18th century, smallpox was endemic worldwide, except in Australia and the small islands of the Pacific. The disease was introduced into Australia in 1789 and in 1829, affecting the Aboriginal population, but eventually declined in both cases [20]. In Europe, variolation was increasingly used as a preventive measure, but probably more people died from its side effects than from natural infection [20].

In some parts of the European countryside, the observation that milkmaids were rarely pockmarked attracted the attention of a British physician. Edward Jenner demonstrated that inoculation of susceptible individuals with cowpox material would protect them from natural smallpox infection, and even challenged these individuals with the inoculation of smallpox virus. Jenner used his own child as a test subject. The results of the vaccinia inoculation were published in 1798, but it was not until 1881 that Louis Pasteur, in honor of Jenner, used the term “vaccination” as a synonym for preventive inoculation against all infectious agents [20].

Smallpox vaccine was the first vaccine ever produced and proved to be a safer method than variolation. The vaccine was introduced in Europe in the early 19th century, and as the incidence of smallpox declined, most governments made smallpox vaccination mandatory. Mass vaccination eliminated smallpox from North and Central America in 1951 and from Europe in 1953, but it remained endemic in the rest of the world [20]. In the 20th century, smallpox still killed between 300 and 500 million people [26].

In 1959, the WHO began a global program to eradicate smallpox. The last case of smallpox in the Western Hemisphere occurred in Brazil in 1971, and the last case worldwide occurred in Somalia in 1977. The WHO declared smallpox eradicated in 1980, 21 years after the start of the global eradication program [20]. Smallpox is the only human disease that has ever been eradicated. Although smallpox is officially eradicated, samples with VARV stains are stored in two laboratories: Centers for Disease Control and Prevention (CDC) in Atlanta (United States) and the Russian State Center for Research on Virology and Biotechnology in Koltsovo, Siberia (Russian Federation), raising concerns about accidents and use of the virus as a biological weapon and subsequent reintroduction of smallpox [26].

## 3. Plague

Plague has afflicted humanity for thousands of years. The disease has killed more than 200 million people [32] and produced the worst pandemic that humanity has ever experienced.

Plague is a vector-borne disease caused by the Gram-negative bacterium *Yersinia pestis* that is transmitted to humans primarily by the bite of the flea *Xenopsylla cheopis*, which infests a wide variety of wild and commensal rodent reservoirs and occasionally humans, although over 80 flea species may also be vectors [32]. Bactrian camels are also hosts for the bacterium and can be infected by flea bites while grazing and resting near rodent burrows. Bactrian camels can transmit the plague to humans if they are ridden or their meat is processed or eaten, a common practice among the Mongolians [33].

Virulence factors of the *Y. pestis* allow it to efficiently infect fleas and subvert the immune system of the mammalian host, leading to a rapid death in the absence of treatment [34]. Long-term antibodies against some antigens of the *Y. pestis* are found in recovered patients even after decades of infection, but the cellular immune response is still unknown [35].

The maintenance of *Y. pestis* in nature is thought to involve enzootic cycles involving rodent reservoirs and fleas that transmit the bacteria at low levels. These enzootic cycles probably pose little risk to humans, but periodic changes in climatic conditions may affect rodent population dynamics and infection may spread to more susceptible rodent species, amplifying the enzootic and causing mass rodent die-offs and immediate dispersal of the infected fleas. The risk of human infection increases when synanthropic rodents such as *Rattus rattus* or *R. norvegicus* and their fleas are involved [36].

The closest bacterium to *Y. pestis* is the enteric pathogen *Y. pseudotuberculosis*, which causes a mild disease [37]. The former diverged from the latter between 5700 and 6000 years ago, by the acquisition of virulence plasmids and inactivation of a virulence-associated gene. The contemporary bacterial lineage diverged from the basal *Y. pestis* lineage between 5000 and 5700 years ago. Approximately 4000 years ago, a new divergence gave rise to an extinct lineage and to lineages that persist to the present day, improving the flea-borne transmissibility and the host invasiveness. However, the geographic origin of this successful lineage has not yet been determined [34].

The most common clinical form of plague is called “bubonic”, in which the infected person suddenly develops high fever, body and abdominal pain, and headache between 3 and 7 days after infection. The bacteria multiply in the lymph nodes closest to the flea bite, causing painful swellings called “buboes”, hence the name “bubonic plague”. Approximately 60% of the untreated individuals die within a week of exposure as the pus-filled buboes suppurate [32]. Rarely, a septicemic or a pneumonic form may occur, accounting for 15% of plague cases [38].

In the septicemic form, the onset of high fever is sudden, with no “buboes” or any other obvious symptoms. The disease progresses rapidly, leading to sepsis, organ failure, and death within a few days. Sometimes patients present with nausea, vomiting, diarrhea and abdominal pain, which can be confused with several other diseases, making the mortality rate usually higher than the bubonic form [36].

Pneumonic plague is the rarest of the clinical forms of plague [32], and is due to the spread of the *Y. pestis* bacterium from a “bubo” to the lungs (referred to as secondary). It begins as an interstitial pulmonary process, resulting in productive cough and abundant sputum, 5–6 days after the infection. If untreated, sputum becomes copious, eventually bloody, and death may occur within 3–4 days. The pneumonic form can also occur when the *Y. pestis* bacterium is inhaled from another pneumonic plague patient or even from an infected animal (referred to as primary). Symptoms are fulminant, and begin within 1–4 days of infection, with sudden onset of fever, chills, headache, malaise, tachypnea, dyspnea, hypoxia, chest pain, cough, and hematoptysis. The sputum is often purulent and copious [36]. In both cases, the *Y. pestis* bacterium is airborne and can be transmitted from person-to-person [32].

### 3.1. Justinian Plague (541–750 AD)

The first plague pandemic is poorly documented, and may in fact have been a process. The first outbreak occurred between 541 and 544 AD, but at least 18 subsequent outbreaks occurred by 750 AD in the Mediterranean, Persia [32,39], and the British Isles [40].

Molecular analysis suggests that it originated in the Tien Shan and Talas Mountains of present-day Kyrgyzstan [40,41]. For this reason, the Huns are thought to have brought the plague to Europe [40]. The first plague outbreak in the Byzantine Empire was reported in Pelusium (Egypt) in 541 AD and reached Constantinople, the capital of the empire, in 542 AD [32]. The spread of the disease was made possible by the maritime trade routes that served the entire empire (Figure 3). Not only were infected rats and fleas inadvertently carried in the ship’s cargo, but so were infected crew members.

It is estimated that between 15 and 100 million people, approximately 25–60% of the population of the Byzantine Empire [39], or approximately 19% of the global population [25], died during this pandemic. The Emperor Justinian the Great, who contracted the disease and survived, was blamed for the plague, as the deaths of peasants led to a collapse of supplies [42] and subsequent famine. These facts were not necessary causes of the fall of the Byzantine Empire, but may have contributed to its vulnerability to foreign invasions.

Although the first pandemic has killed tens of millions of people, it bears little resemblance to the next [39], which could be defined as the single stochastic event that brought humanity the closest to an extinction bottleneck. This may be related to the evolution of the *Y. pestis* lineages. A divergence event in the Tien Shan and Talas Mountains of present-day Kyrgyzstan [40,41], just before the first pandemic gave rise to the Justinian lineage (which went extinct immediately after the pandemic) and new lineages associated with the subsequent pandemics [34].

### 3.2. Black Death (1338–1353 AD)

A mortality crisis between 1338 and 1339 AD in the Chu Valley, near the Lake Issyk-Kul’ in present-day Kyrgyzstan, may be the earliest evidence for the origin of the Black Death. This evidence comes from an epigraphic corpus of gravestone inscriptions from three cemeteries in the Issyk-Kul’ region [33]. Further paleogenetic analysis of two of these cemeteries provided the definitive link between the Issyk-Kul’ mortality crisis and the Black Death [43]. Conversely, the peak mortality and the demographics of the dead were similar to the subsequent Black Death, with significantly more males affected [33].

Sporadic mass mortality events among Mongol soldiers in the early 14th century, during the incursions into China, and three outbreaks in southeastern China in 1333 AD, between 1344 AD and 1345 AD, and in northern China in 1353 AD, have been referred to since the 19th century as the “Chinese origin” thesis of the Black Death [44], but these localized outbreaks were actually unrelated to the pandemic [33].

In the early 14th century, the Central Asian landscape was largely pastoral, providing the Mongols with horses, livestock and Bactrian camels. Pastoralism was directly correlated with rainfall levels, which in turn were directly correlated with the abundance of rodents and livestock [33]. The first half of the 14th century was characterized by intermittent droughts and floods. For example, the year 1339 AD was the driest year and 1343 AD was the wettest year of the period [45]. The drought may have reduced vegetation to the point where it could barely support the rodent population, which in turn may have increased mortality and decreased both fertility and immune response, making the surviving rodents increasingly susceptible to flea infestation and *Y. pestis* infection. This condition may have led to spillover of the bacterium from rodents to humans, while the fleas actively sought alternative hosts for survival [33].

Issyk-Kul’ was an important hub of international trade at that time. Spices, silk, cotton, linen, and jewelry traveled westward. Silk, cotton, and linen clothing may have been used by the fleas as hiding places until they jumped onto a human host. *Y. pestis* can survive in clothing for long periods of time without the need for live fleas. Infected fleas could also be transported in grain or flour. Commensal rats infected with the bacterium or infested with fleas may have been transported on cargo ships that allowed the rats to disembark through the mooring ropes when docking at destination ports. Although Bactrian camels are hosts for *Y. pestis*, they were rare in Central Asia and may have played a minor role in the spread of the plague from Issyk-Kul’ [33].

In late 1346 AD, while the Mongol army was besieging the city of Caffa, Crimea, the soldiers began to die. Their bodies were then catapulted into the city in an attempt to create terror and also to infect the city’s population [33]. Genoese merchants, fleeing war and death, imported the plague to Constantinople in May and to Messina (Sicily, Italy) in June of 1347 AD. Wine ships from Gascony (southwestern France) brought the disease to Borset (southwestern England) in June 1348 AD. Merchant ships from England introduced the disease to Scotland, Shetland, the Faroe Island, and Bergen, Norway [33]. From the coast, the disease spread inland, terrifying citizens who saw their relatives, friends, and neighbors die within a week of contracting the disease. The plague spread rapidly to North Africa by 1349 AD, to all Europe by 1350 AD, and to the Middle East by 1351 AD [32] (Figure 4).

In 1377 AD, Ragusa (nowadays Dubrovnik, Croatia) established that suspected plague cases should spend thirty days in a landing station far from the city, a period referred to as “trentena”. In 1403 AD, the “trentena” was extended (mostly for biblical reasons) to 40 day (“quaranta giorni”) isolation, referred to as “quarantena” from which the term quarantine derived from [46].

The impact of the Black Death was enormous, changing European society and economies in many ways, including a severe labor shortage, the decline of the Italian city-states, and the Christian belief that the plague was carried by Jews, which led to persecution and massacres. In addition, the miasma theory proposed by the ancient physicians Hippocrates (460–377 BC) and Galen (129-c. 216 AD), which attributed disease to poisoned air, gained relevance [32].

This pandemic heavily massacred the European population, already devastated by famine and war, and became the deadliest pandemic in human history. The Black Death is thought to have killed between 75 and 200 million people, approximately 30–60% of the European population [47] or up to 51% of the world’s population [25]. Despite recent research suggesting that the death toll may have been lower [48], by 1430 AD, the population of Europe was lower than it had been in 1290 AD, and it was not until the 16th century that pre-Black Death numbers were recovered [46]. The Black Death was followed by successive plague outbreaks over the centuries until the next pandemic broke out [38].

### 3.3. Third Plague (1855–1960 AD)

The third plague pandemic began in the Chinese province of Yunnan. The region had experienced several plague outbreaks since 1772 AD [49], but this pandemic began in 1855 AD [40]. After reaching Canton and Hong Kong in 1894 AD, the disease was spread by rat-infested steamships to port cities in Japan, Singapore, Taiwan, and India. After a few years, the disease spread to several other port cities around the world (Figure 5).

The plague bacterium was identified in Hong Kong in 1894 AD, by the Swiss-French physician and bacteriologist, Alexander Yersin [49]. Later, the plague bacterium was named in his honor, *Y. pestis*. In addition, in 1897, the French scientist Paul-Louis Simond identified the *Y. pestis* bacterium in dead rats while studying the plague in Bombay, India. He proposed that the bacterium was transmitted between rats and from rats to humans [38].

Following these discoveries, international regulations for rat control in ports and inspection of ships were established in the early 20th century. However, the *Y. pestis* established in urban and rural rodent hosts plague-free areas in Americas, Europe, Africa and Asia, resulting in scattered outbreaks that still exist in some parts of the world [38].

Deaths were concentrated in India, where mitigation measures were instituted by the British colonial government imposed mitigation measures such as quarantines, isolation camps, traffic restrictions, and bans on traditional medical techniques, considered leonine by the local population, leading to riots [50].

The enforcement of sanitary standards and, from the mid-20th century on, and the availability of effective insecticides, rodenticides, and antibiotics for treatment, made the plague sporadic and rural. Outbreaks developed more slowly and were easier to control [38]. By 1960 AD, the plague had caused at least 15 million deaths worldwide [46], or at least 1% of the world’s population [25], with the majority occurring in India [46]. 

However, scattered outbreaks of plague still occurred in Vietnam, during the 1962–1975 war [38], China and Tanzania in 1983 AD, Zaire in 1992 AD, and India, Mozambique and Zimbabwe in 1994 [46].

In 1892, a vaccine against the plague was developed by a Russian scientist named Waldemar Mordecai Haffkine, while working at the Pasteur Institute in Paris. The same researcher developed the first vaccine against a bacterium, *Vibrio cholera*, in 1882, against cholera (see below). Clinical trials for the plague vaccine were conducted in India from 1893 to 1896 in India, using prisoners and Haffkine himself as test subjects [51]. When the plague reached Bombay, in 1896, Haffkine became a bacteriologist for the British colonial government of India, and his effective vaccine was used in a major vaccination campaign. In 1902, however, 19 people died of tetanus as a result of vaccine poisoning. Haffkine was fired, but the vaccine stigma remained in India [51].

## 4. Cholera

Cholera has plagued humanity for thousands of years. Although the origin of the disease is unknown, the first records of a cholera-like disease, in the Ganges River delta, are found in Sanskrit medical texts dating from 500 to 400 BC [52]. Historically, and especially during the 19th century, the Ganges river was known for its polluted waters, where corpses were thrown. Moreover, along its course, the lack of hygienic conditions and the use of domestic water reservoirs exposed to air, excrement and urine, created the perfect conditions for a waterborne pathogen to thrive [53].

Cholera is a severe and dehydrating diarrheal disease that is considered one of the most rapidly fatal human infections if not treated promptly. It is a waterborne disease caused by the Gram-negative aquatic bacterium *Vibrio cholerae*. The bacterium is motile, and occurs naturally in marine environments, using the chitin of crustaceans as a source of carbon and nitrogen [52]. Over 200 serogroups of *V. cholerae* have been identified, based on its lipopolysaccharides. Serogroups, O1 and O139, are those associated with epidemics and are the only serogroups that secrete the cholera toxin. Serogroup O1 is subdivided into the classical biotype and the El Tor biotype. The first six cholera pandemics were caused by the classical O1 biotype, and the ongoing seventh pandemic is caused by the El Tor biotype [54].

The *V. cholerae* is transmitted by the fecal–oral route. A high infectious dose (10^8^ vibrios) is required to cause disease. Human-to-human transmission is possible and well documented. *V. cholerae* is present in human stool both as individual vibrios and as biofilm aggregates. Cholera outbreaks are associated with water sources, natural disasters (such as floods, hurricanes and earthquakes), and host immunity. The bacterium enters the host through contaminated food or water. In the small intestine, it invades the epithelial cells, multiplies and secretes cholera toxin, resulting in an efflux of ions and water into the intestinal lumen. The resorptive capacity of the colon may be overwhelmed, resulting in profuse and severe diarrhea [52]. Taken together, these factors lead to the association of cholera with poor sanitation, poor food hygiene and poverty [55].

Clinical signs are variable, ranging from asymptomatic to mild to severe. The incubation period is between 0.5 and 5 days, after which there is a sudden onset of diarrhea and often vomiting. In severe cases, the stool output can reach 1000 mL/h in an adult patient. If untreated, the disease can lead to severe dehydration, hypovolemic shock. And death [52]. Without treatment, the case fatality rate can exceed 50% [54].

Asymptomatic patients shed the bacterium for a few days, but symptomatic patients may shed it for up to 2 weeks or even longer. Cholera infection protects against subsequent serogroup homologous infection for at least 3 years [54]. Therefore, less severe disease is observed in cholera-endemic areas [52].

It is difficult to distinguish between the first six cholera pandemics because they were separated by only a few years. Nevertheless, many authors divide them in one way or another, following the description of Pollitzer [56]. Seven cholera pandemics have occurred in history, killing up to 40 million people [11].

### 4.1. First Cholera Pandemic (1817–1824 AD)

Cholera had long been endemic in some regions of South Asia when the first pandemic began. Some cases had been recorded in India since 1814, but in 1817, 10,000 people died in a village near Calcutta. Moreover, British soldiers stationed in Calcutta also fell ill, and the disease spread rapidly across the Indian subcontinent in the following year [53]. The disease spread to the north to China and Nepal, east to what is now Sri Lanka, Myanmar, Mauritius, Thailand, Malaysia, Singapore, the Dutch East Indies (now Indonesia), the Philippines and Japan, and west to Zanzibar (Africa), the Middle East, Russia, Georgia, Syria, Egypt, and the Mediterranean coast in 1823 [57] (Figure 6).

Colonialism, i.e., European colonial rule, and globalization, i.e., the increased speed of worldwide transportation (especially by steamship), have been cited as the causes of the spread of cholera beyond the Indian subcontinent [13]. Conversely, the reluctance of British Indians to accept that cholera was communicable led to the disease being associated with climate or miasma. Elsewhere, on the other hand, quarantines and isolation of victims were instituted. Even when traditional treatments were introduced, the course of the disease was so rapid that there was no time for them to have any effect (good or bad) [13].

The exact number of deaths from the first cholera pandemic is unknown, but estimates vary from 1 million [11] to 2 million based on local reports, although several authors have reported much higher numbers [13,53,57], accounting for up to 0.3% of the world’s population [25]. It is believed that this pandemic is poorly described because it did not reach Europe, despite the European fears of the so-called “Asian cholera” [13].

### 4.2. Second Cholera Pandemic (1826–1835 AD)

In less than two years after the remission of the first pandemic, cholera began to spread again in the Ganges River delta region between 1826 and 1827. During the same period, the disease was reintroduced into China [53,56]. Like the first pandemic, this one reached Persia and Afghanistan in 1829 [13], Japan in 1831 [56], and Java in 1834 [13]. On the other hand, it first reached the Russian Empire in 1830, Western Europe in 1831, and North America, in 1832. Muslim pilgrims brought the disease from Mecca to Palestine, Syria, and Egypt in 1831. Pilgrims carried the pandemic even further into East Africa (Sudan, Abyssinia, the Horn of Africa, and Zanzibar) [13] and Tunisia by 1835 [56,57] (Figure 7).

The impact of the pandemic was more dramatic in Europe, but in Asia, the effects of the second pandemic were less severe than the first pandemic. The significance of this pandemic has more to do with the social and political situation in Europe than with the mortality itself. This pandemic highlighted poverty and unsatisfactory sanitary conditions, leading to social tensions and to attacks on the authorities [13]. At least 1 million people died in this pandemic [53,58], or 0.2% of the world’s population [25].

### 4.3. Third Cholera Pandemic (1839–1860 AD)

The third cholera pandemic also began in the Bengal region of India, less than four years after the previous pandemic. Cholera has been endemic in India, and some authors disagree about when the second pandemic ended and the third began, but there was a pause in the number of cholera cases in India between 1836 and 1839 AD, when a new surge began [13].

This pandemic spread almost worldwide, reaching Latin America for the first time, and returning to Europe and North America. The disease spread westward, to Bombay in 1839 and from there, to Aden in 1846, and to the Red Sea region and North Africa in 1848. Again, the pilgrim traffic to and from Mecca was associated with the spread of cholera in the region. Also from Bombay, another branch of the disease spread to the Persian Gulf region in 1846, and from there, the Russian Empire via the Caspian Sea in 1847, and to Western Europe the following year [13]. The Crimean War helped spread cholera to the Mediterranean ports and the Balkans [53]. From Europe, the disease reached North and Central America in 1849 and South America for the first time in 1855. Apart from India, the highest mortality was observed in Brazil [13,57].

From the Bengal region to the east, the disease reached Malaysia, Singapore, and southeastern China in 1840, probably carried by British troops and trade during the Opium War between China and the United Kingdom. From the coast, the disease spread inland, reaching Burma, western China and from there, Central Asia in 1844 [13] (Figure 8).

The third cholera pandemic killed between 1 and 1.5 million people [13,53], or 0.08 to 0.12% of the world’s population [25]. In some places, such as Brazil, the Caribbean, parts of the United States, and Europe, the mortality rate was high, making the third cholera pandemic the worst of the 19th century. The mortality rate was associated with slavery and killed mostly men in Brazil and the Caribbean. Although the mortality rate of the third cholera pandemic was higher than that of the second, the former was not accompanied by social unrest [13].

The British physician John Snow suspected that cholera was waterborne ever since the second cholera pandemic hit London in 1834. Although cholera struck London again in 1848, it was not until 1854, when the number of deaths peaked, that Dr. Snow began to investigate the outbreak in the Soho district by plotting the deaths on a map. The deaths were clustered around a particular water pump on Broad Street (now Broadwick Street), and Dr. Snow hypothesized that the contaminated water from this pump was the source of the outbreak. Although the number of cholera cases declined, Dr. Snow convinced the local authorities to remove the handle of the pump, which further reduced the number of cases. Proving that cholera was waterborne was a major breakthrough in understanding its transmission. This work is the earliest example of epidemiological investigation, and for this reason Dr. Snow is referred to as the father of epidemiology [13,53].

### 4.4. Forth Cholera Pandemic (1863–1875 AD)

The fourth cholera pandemic began in 1863, like the previous ones, in the Ganges River delta region. The following year, the disease reached Bombay, on the west coast of India, and from there spread around the world, following the path of previous cholera pandemics, but with a difference. From India, the disease reached the port of the British protectorate of Aden in 1864. From Aden, the disease spread to Somalia and Ethiopia in 1865 and, more importantly, to Jidda, the seaport of Mecca, where Muslim pilgrims from all over the world were gathered. That year, approximately 17% of the 90,000 pilgrims died of cholera. From Egypt, the disease spread across the Mediterranean Sea. In 1865, cholera reached Great Britain, the Netherlands, Prussia and Russia. From Europe, the disease was transported to the West Indies and North America, in 1866. From Aden, cholera spread not only to Egypt, but also to Algeria, Tunisia and Morocco in 1867 and to Senegal in 1868. From Ethiopia, the disease spread across the African continent to Zanzibar, Mozambique, and Madagascar in 1869 [13,57] (Figure 9).

The fourth cholera pandemic killed approximately 700,000 people worldwide [57], or 0.06% of the world’s population [25]. The pandemic was more severe in India, especially when it coincided with a year of famine, when droughts forced the people to drink from contaminated wells. Famine was associated with rural-urban migration and consequently poor water supply and sanitation [13].

This pandemic was particularly associated with religious pilgrimages by Western societies, and Western empires began to regulate these pilgrimages. It was also the first time that cholera was associated with human movement, being imported from somewhere else. The belief that cholera was associated with water pollution grew, but it was not until the next cholera pandemic that the pathogen was discovered [13].

### 4.5. Fifth Cholera Pandemic (1881–1896 AD)

In 1881, cholera spread around the world for the fifth time, again starting in the Ganges River delta area, again. In 1882, the pandemic spread outside India to the Dutch East Indies (now Indonesia) and the Philippines, causing high mortality in both places. The following year, the disease again reached the Muslim pilgrims in the Hijaz and Mecca regions of Arabia, and from there to Egypt. In 1884, the disease moved through the Mediterranean ports and spread to Italy, France and Spain. In 1886, the pandemic reached Argentina, and from there it spread through South America the following year. Between 1888 and 1891, the disease retreated to Asia, causing new outbreaks, but reappeared in Europe in 1892, using the same route as the second cholera pandemic, up the Volga River into the Russian Empire. From the Russian Empire, the disease spread to Germany and to Western Europe [13,57] (Figure 10).

The fifth cholera pandemic killed approximately 750,000 people worldwide [57], or 0.06% of the world’s population [25]. The pandemic hit India severely, and especially affected the Russian Empire, causing social unrest due to the high mortality rate [13].

In 1854, the Italian doctor Filippo Pacini described an active vibrio in stool samples from cholera patients in the city of Florence, Italy. The discovery of *Vibrio cholerae*, which called into question the theory of miasmas, was ignored by the scientific community of the time. The causative agent of cholera only came to prominence in 1883, when Robert Koch, who was already a renowned microbiologist, managed to isolate the bacterium during a visit to India during a cholera outbreak [59]. In addition, between 1889 and 1896, at the end of the pandemic, a clinical trial carried out by the Russian scientist Waldemar Haffkine with approximately 40,000 volunteers in Calcutta, India, made it possible for the first time to use an attenuated human vaccine against cholera [57]. Unfortunately, even the existence of a vaccine was not enough to prevent a new cholera pandemic.

### 4.6. Sixth Cholera Pandemic (1899–1923 AD)

The sixth cholera pandemic began in 1899 in the same place as the previous five: the Ganges River delta. The disease had long been endemic in India, but this would be the last time a cholera pandemic broke out in this location. As usual, the disease spread westward to Bombay, Madras, and Punjab, and from there to other countries. In 1901, the disease reached Singapore and the Dutch East Indies (now Indonesia) from Indian ports. In 1902, it reached China and the Philippines, Muslim pilgrims in the Hijaz, and eventually Egypt. The disease reached Persia in 1903 and the Russian Empire in 1904. In 1905, cholera reached the Austro-Hungarian Empire [13,57].

The disease seemed to disappear between 1905 and 1909, but reappeared in some places such as India, the Dutch East Indies, and the Russian Empire. The disease reached Western Europe in 1910, particularly Italy and the Balkans. From Italy, it reached the Ottoman colony of Tripolitania in 1911, probably brought by the Italian soldiers occupying that territory [13] (Figure 11).

The First World War and the Russian Revolution, whose ensuing civil war lasted until 1921, greatly facilitated the spread of cholera, especially in Russia and on the Eastern and Southeastern European fronts. This cholera pandemic occurred at a troubled time in history, during the World War I and the most severe influenza pandemic of 1918–1919, the Spanish flu. Even so, as in the previous cholera pandemic, the United States had cases, but they were contained by quarantines of passengers on ships coming from affected countries [13]. 

The sixth cholera pandemic caused a large number of deaths, particularly in India, where approximately 8 million people may have died from cholera [13]. Outside of India, cholera killed approximately 800,000 people worldwide [57]. The total death toll of the sixth cholera pandemic was approximately 8.8 million people, or 0.6% of the world’s population [25].

In the early 20th century, cholera was responsible for 10% of deaths in India. In the Philippines, cholera was responsible for nearly 1/3 of deaths. In Persia (now Iran), cholera caused social disruption due to the high mortality rate [13]. With the growing consensus on the cause of the disease, some governments began to adopt public policies to control cholera, but without success due to the difficulty of adapting scientific knowledge to the reality of societies [13].

### 4.7. Seventh Cholera Pandemic (1961 AD–Today)

This time, the cholera pandemic began in Southwest Asia in 1961, for the first time outside the Ganges River delta. The seventh cholera pandemic continues to this day and has so far been bimodal, i.e., with cases and deaths accumulating at two different times: from 1970 to 1980 and after 1991 [13].

The seventh cholera pandemic began on the island of Celebes, near the city of Makassar, Indonesia. From there, it spread throughout Southeast Asia, including the Philippines and Hong Kong, in 1961. The following year, it reached Borneo (and the rest of Indonesia) and Taiwan. In 1963, it reached Korea, Burma and Bangladesh. Between 1964 and 1965, the disease spread westward, reaching India, Iran, Iraq and Bahrain. After this breakthrough, the pandemic did not advance further geographically until 1970, when it expanded dramatically into Africa and Europe, using two previously used routes: (a) into the Soviet Union via the Caspian Sea to Astrakhan and Odessa on the Black Sea, (b) via the Arabian Peninsula to the Middle East and East Africa, and a new route, via air transport to West Africa, specifically Guinea. The following year, the disease spread to East and Southwest Africa. Europe was also hit that year, especially Portugal and Spain, and then the entire southern part of the continent. The main European country affected was Italy in 1973. Muslim pilgrims were affected again in 1974, hitting the countries of origin as the pilgrims returned to their countries of origin [13,57].

Cholera took on an endemic pattern from the late 1970s until 1991, when it hit Peru and, from then on, almost all of Latin America. As early as 1991 (the year the disease was introduced), cholera cases in Latin America accounted for two-thirds of the world’s cases [13] (Figure 12).

The exact number of deaths during this pandemic is controversial, mainly because the different countries did not have organized epidemiological surveillance services from the beginning of the pandemic. Nevertheless, the WHO counted approximately 235,000 deaths from cholera between 1961 and 1989 [60], and Ilic and Ilic [61] estimated the total number of deaths between 1990 and 2019 at approximately 3 million or 0.1% of the world’s population [25].

The reasons for the spatio-temporal dimension of this pandemic may be related to the appearance of a milder biotype of *V. cholerae* (El Tor), which allows for rapid spread and the generation of undetectable cases without causing panic in the population due to the low perception of a crisis. In addition, air travel and labor migration meant that, even for one of the diseases with the fastest known clinical course, patients could be transported from one place to another without dying and then transmitting the disease to new susceptible people. Currently, the disease has been controlled through the use of vaccines, antibiotics and fluid replacement, resulting in the lowest mortality rate of any cholera pandemic [13,57].

## 5. Influenza

The first reference to influenza is found in 412 BC, in Hippocrates’ sixth book on “Epidemics” [62]. However, the term influenza was not proposed until 1357, in Florence, as “influenza di fredo”, from the Italian “influence of the cold” [12].

Influenza is a highly contagious airborne disease that commonly occurs in winter and eventually causes seasonal epidemics. Although the virus is mainly transmitted directly by aerosols it can also be transmitted by fomites [63]. The disease manifests itself as an acute fever with varying degrees of respiratory involvement. Other symptoms include, in addition to fever, chills, headache, weakness, red eyes, sore throat, dry cough and nasal discharge. The disease is caused by the influenza virus, which belongs to the Orthomyxoviridae family. The Orthomyxoviridae family consists of nine genera, four of which are involved in in vertebrate influenza [64]. Influenza viruses are enveloped with a genome consisting of segmented, negative single-stranded RNA. The four genera, or types, of vertebrate influenza are differentiated by antigenic combinations of two proteins encoded by the viral genome: hemagglutinin and neuraminidase. The antigenic combinations of these proteins (16 for hemagglutinin and 9 for neuraminidase) determine the four known types of the virus: A (IAV or α-influenzavirus), B (IBV or β-influenzavirus), C (ICV or γ-influenzavirus) and D IDV or δ-influenzavirus). Type A is the most common and causes mild to severe seasonal illness in humans and animals and is involved in the vast majority of outbreaks and pandemics. Type B is highly contagious and can cause severe disease in humans and pinnipeds. Type C primarily and sporadically infects humans, but can also infect pigs. Finally, type D primarily infects pigs and cattle and has not been detected in humans [63,65]. It is currently believed that influenza viruses originated in sturgeon fish, and that these animals were the first hosts of Orthomixoviridae [66].

The virus evolves rapidly through antigenic variation, through mutations resulting from antigenic drift and shift. These new variants show differences in the expression of hemagglutinin and neuraminidase. Antigenic shift is responsible for influenza pandemics, mainly associated with type A [65]. Anseriformes (aquatic waterfowl) can carry type A influenza viruses with all the hemagglutinin subtypes (H1-H16) and neuraminidases (N1-N9) and are therefore considered natural reservoirs of the virus [65]. As a consequence, human epidemics have been caused by avian viruses: H5, H7 and H9.

After infection, symptoms begin between 18 and 72 h and the virus begins to be transmitted 24 to 48 h after the onset of symptoms. Symptoms can last for up to seven days, but fever lasts only three days [65]. Complications of the viral infection include secondary bacterial infections, mainly in the form of pneumonia, which significantly increase mortality [63]. Mortality from influenza can be determined by the excess of deaths in non-pandemic years, which has been estimated at 9.1 per 100,000 in the United States, or approximately 0.001% of registered deaths, with high seasonal variation [67], and 0.004–0.009% worldwide [68]. In interpandemic years, approximately 90% of deaths occur in individuals over 65 years of age [68].

Seasonal influenza kills between 295,000 and 518,000 people worldwide each year [69], whereas 15–25% of the population is infected [70]. The first influenza pandemic is thought to have occurred between 1557 and 1580, affecting the entire world, persisting in the United Kingdom and Europe [71], but the involvement of the influenza virus and the number of deaths have not yet been confirmed. At least six confirmed influenza pandemics have occurred in history, killing between 25 and 113 million people [72,73,74,75,76].

### 5.1. Russian Flu (1889–1890 AD)

The first documented influenza pandemic began in late 1889 in a city in the south of the Russian Empire, Bukhara (now Turkmenistan) [77]. Using railroads and waterways, the disease spread rapidly westward, reaching Moscow and the empire’s capital, St. Petersburg [13]. The increase in hospitalizations caused the health system to collapse, and even Tsar Alexander III was infected [77]. From the Russian Empire, the disease spread in approximately a month to the major European cities, especially Paris, Berlin and Vienna. By the end of 1889, railroad transportation was largely responsible for the spread of influenza throughout the European continent, except for a few islands (the United Kingdom and Sardinia). Maritime transportation brought the disease to the United States in December 1889, and it spread to the Midwest in January 1890 [13].

Passenger transportation, both by rail and especially by steamship, was the primary vector for the introduction of the influenza virus, and where this transportation network was less dense or absent, the rate of spread was slower. By December 1889, the disease had spread to Mediterranean ports and, by January 1890, to ports in Africa, South America, Japan and the United States. The following month, more African ports were affected, as well as Hong Kong and Singapore, and in March, Bombay, and India [13] (Figure 13).

Almost everywhere, the epidemic curve was the same: approximately 15 days after the first cases, the peak incidence was reached, which lasted for up to three weeks, and then, a month after the first cases, it suddenly declined due to the lack of susceptible individuals. The disease affected all ages, sexes, and social classes without discrimination. During the epidemic period, the morbidity rate was extremely high worldwide [13]. The number of deaths is estimated at between 3.7 and 5.1 million worldwide [76], or 0.2 to 0.3% of the world’s population [25]. It is also estimated that one-third to one-half of the world’s population may have been infected during this period. Although the usual mortality rate from influenza is very low, the large number of people infected led to a large number of deaths. Although the pandemic ended quickly in the mid-1890s, some isolated outbreaks occurred in cities until 1893, mostly affecting the elderly [13].

The influenza virus that may have been involved in this pandemic was the A-H3N8, which also causes epizootics in horses [12]. In 1889, there was an epidemic of equine influenza in the United Kingdom. The hypothesis of H3N8 as the cause of Russian flu was based on serologic studies, which provide only indirect evidence and are subject to technical artifacts and cross-reactions [77]. There is also a conflicting speculation that the pandemic was caused by the HCoV-OC43 coronavirus, which, along with other coronaviruses, is currently responsible for up to 30% of cases of the common cold. HCoV-OC43 is similar to the bovine coronavirus (BCoV), from which it was derived around 1890. Workers may have been exposed to bovine respiratory viruses during the eradication of a panzootic disease thought to be caused by *Mycoplasma mycoides* peripneumonia between 1870 and 1890. It was observed that in 1889, cattle showed the same signs as humans affected by Russian flu. Thus, the hypothesis that the Russian flu was caused by a coronavirus rather than influenza is based on the assumption that a mutant coronavirus emerged from a bovine coronavirus as a result of exposure of humans in direct contact with cattle [77]. However, this hypothesis remains unproven.

### 5.2. Spanish Flu (1918–1920 AD)

One of the most devastating pandemics humans have has ever faced, the Spanish flu, began in early 1918 and lasted until early 1919. The name given to the pandemic does not reflect its origin, but rather the fact that Spain, by staying out of the First World War (between 1914 and 1918), allowed the press to report on the disease, without the having to hide it from its enemies [13]. RNA was extracted for molecular characterization from formalin-fixed lung tissue samples and unfixed and frozen samples from influenza victims buried in the Alaskan permafrost in 1918 and found to be compatible with the influenza A-H1N1 virus [78].

This pandemic was characterized by three epidemic waves [13]. The first wave began in March 1918 in the hospital ward of the U.S. Army training camp called Camp Funston, in Kansas, United States [79], and spread to Western Europe in April, with the arrival of American troops entering World War I, to China and India in May, and to Northern Europe, Oceania and Southwest Asia in June. At that time, Germany planned to launch a major offensive on the Western Front, and found the timing perfect with the collapse of the Soviets on the Eastern Front and the beginning of the American offensive in the West. The Germans failed precisely because of an influenza outbreak in June that affected approximately half a million soldiers. The German offensive ended in June, just as the first wave of the pandemic ended [13].

The second wave began in France in August 1918 and spread to Boston, United States, and to the coast of West Africa. By the following month, it had already spread to the west coast of the United States, virtually all of Europe and West Africa. By October, South and Southwest Asia were affected, and by November, Siberia and the Pacific Islands. Very few places were unaffected, such as Iceland and some Pacific islands. This wave was characterized by a change in the clinical pattern of influenza: a high incidence of pulmonary complications, such as pneumonia, leading to high mortality. In the United States and Europe, the mortality rate was approximately 5 per 1000, but in some regions of the world, particularly Africa and Asia, it was much higher. Another change was the age group affected, mainly young adults, who have been the most resilient age group in other pandemics. Nevertheless, the pattern of the epidemic curve remained the same, with a rapid increase in incidence, a peak lasting up to three weeks, and a rapid decline, due to the lack of susceptible individuals. The end of the second wave coincided with the end of World War I, in November 1918 [13].

Between November and December 1918, the incidence of influenza had declined, but increased again between January and February 1919, beginning the third wave of the pandemic. However, this wave was not as severe as the previous one and quickly subsided [13] (Figure 14).

The H1N1 virus was uniquely virulent, and the number of deaths has been estimated at 17.4 to 100 million worldwide [72,73,76], or 0.95 to 5.4% of the global population [25], and approximately one-third of the world’s population was infected [80]. As in other influenza pandemics, a significant proportion of the population was infected with the influenza virus, resulting in high morbidity worldwide [13].

The severity of the pandemic led to the adoption of restrictive preventive measures, mainly in the United States and Europe, such as the prohibition of public gatherings (closing of schools, churches, theaters, etc.) and the use of masks covering the mouth in public places. Such measures were met with varying degrees of acceptance by populations and governments. Governments have also had to deal with the collapse of health services. In addition, several essential services (garbage collection, undertakers, gravediggers, etc.) were affected by the demand for military conscription. In countries under European imperial control, the same preventive measures were not taken, due to the war effort, and influenza spread more rapidly [13]. After the pandemic, the influenza virus returned to its usual pattern of virulence and endemicity, until the next pandemic [81].

### 5.3. Asian Flu (1957–1958 AD)

Since the first isolation of a human virus occurred in 1933, this would be the first influenza pandemic to be studied in the laboratory, beginning with the characterization of a new influenza A virus, H2N2 [81]. The H2N2 virus was the result of a mixture of genetic material from different influenza viruses (human H1N1 with avian H2N2) generated in pigs. Pigs are susceptible to both human and avian influenza viruses, allowing the exchange of genetic material from these strains and the generation of new viruses that are more easily transmitted to humans [82] and between humans. A new influenza virus would once again challenge the world’s population, 38 years after the last pandemic [81].

In 1957, even with a surveillance system less prepared than today’s, researchers from Australia, the United Kingdom and the United States had already isolated the new virus shortly after recognizing a severe influenza outbreak in Hong Kong [81]. In fact, the first wave of the pandemic began in February 1957 in Guizhou Province in southwestern China and spread to Hong Kong in April. From there, it spread to Japan and Singapore in May [83]. By the end of May, the disease had already spread to Southwest Asia, Indonesia, the Philippines [84] and Taiwan [85]. In June, the disease reached India, the United Kingdom and the United States [85] (Figure 15). The second wave, which occurred between January and February 1958, primarily affected the elderly and was therefore more deadly than the first wave.

During the Spanish flu, secondary bacterial lung infections were the main cause of high mortality, but during the Asian flu, sudden death with consolidation or pulmonary edema without secondary bacterial infection was more common. Also differently, those with chronic lung or heart disease were most affected [81]. The number of deaths during the Asian flu pandemic has been estimated at 1.1 to 2.7 million worldwide [76,86], or 0.04 to 0.1% of the global population [25].

In 1942, an influenza vaccine was developed by the U.S. Armed Forces [87] and introduced to U.S. troops in 1946, with a protocol of annual revaccination beginning in 1954. This was the first time that the response to vaccination of a population exposed to a new influenza strain could be observed. An increase in population antibody levels in the population was observed in the years following the pandemic, demonstrating a good response to vaccination [81]. The post-pandemic circulation of H2N2 in the human population was short, and it was replaced by a new strain of the influenza virus in the next pandemic 11 years later.

### 5.4. Hong Kong Flu (1968–1969 AD)

A new influenza pandemic began in July 1968, again in Southeast Asia. Although its exact origin has not been determined, it was dubbed the Hong Kong flu because the influenza epidemic in this place had attracted attention, especially from the West [81]. This time, the influenza virus involved was A-H3N2, also the result of mixing the genetic material from different influenza viruses using pigs as an intermediate host to optimize for human transmission [82].

In August, it struck Singapore, the Philippines, Taiwan, Vietnam and Malaysia. In September, Thailand, India, Australia and Iran were affected. In Japan, despite several introductions, some outbreaks were reported in October, but the epidemic was postponed until January 1969. In October, the disease reached California, United States, and by November it had reached the East Coast. By early 1969, all of Europe was affected. Some tropical countries were not hit in the first wave, but in the second wave by early 1969, such as Brazil, Sri Lanka, and Indonesia. By mid-1969, the entire Southern Hemisphere was affected by a milder form of influenza [88] (Figure 16).

As in previous influenza pandemics, the peak of deaths occurred two weeks after the peak of the epidemic curve, and was as severe as the 1957–1958 pandemic. The number of deaths during the Hong Kong flu pandemic has been estimated at 2 to 3.8 million worldwide [76], or 0.06 to 0.10% of the global population [25].

### 5.5. Russian Flu (1977–1979 AD)

An influenza epidemic that began in the Soviet Union in November 1977 caught the attention of health authorities. However, it was later discovered that the cases began in May in northeastern China. It attracted attention because the age group affected was under 25, making it a mild disease. The fact that this age group was affected was attributed to the possible involvement of the H3N2 influenza virus, which had replaced H2N2 after 1968, which in turn had replaced H1N1 in 1957. However, the characterization of the 1977 virus showed that the antigens were compatible with the H1N1 virus circulating in the 1950s [81].

This finding was troubling, because antigenic drift would have significantly altered the virus in two decades of circulation in humans. The theory of reactivation of a dormant strain was dismissed as incompatible with the biology of influenza viruses, so it was theorized that samples of the influenza virus from the 1950s had been frozen and kept for in vivo experiments, possibly for vaccine development. The virus then escaped and entered the population. However, these theories have not been confirmed to date [81].

### 5.6. Swine Flu (2009–2010 AD)

In February 2009, a new flu pandemic began in the city of La Gloria, Veracruz State, Mexico. The disease spread rapidly around the world through air travel [89]. By December 2009, 208 countries and territories had reported at least one laboratory-confirmed case (Figure 17). The influenza virus involved was characterized as H1N1 and was derived from two unrelated swine influenza viruses, one of which was derived from the 1918 human H1N1 influenza virus [90].

H1N1 has been endemic in pigs since the Spanish Flu pandemic, having been first isolated in the United States in 1930. This virus has been transmitted to humans on several occasions, with a mortality rate of approximately 17%, especially in pregnant women and immunosuppressed individuals in epidemics resulting from direct contact with pigs [90]. However, since 1996, some influenza viruses (H7N7, H5N1 and H9N2) have been transmitted directly from birds to humans, without the need for an intermediate host for optimization, i.e., pigs, but there is currently no human-to-human transmission of these viruses [82].

No significant changes in transmission and clinical characteristics were observed with respect to seasonal influenza, i.e., most patients had a mild illness and recovered spontaneously. In the swine flu pandemic, most of these patients were adolescents and young adults. It is estimated that of the total number of infected people, between 1 and 10% were hospitalized, of which 10 to 25% required intensive care and, of which 2 to 9% died. It is believed that people under the age of 30 did not have enough protective antibodies to deal with the H1N1 infection, while people over the age of 50 were exposed to the H1N1 strains circulating before 1957 [90].

No significant changes in transmission and clinical characteristics were observed in relation to seasonal influenza, i.e., most patients had a mild illness and recovered spontaneously. However, most of the hospitalized patients have some medical conditions such as asthma, diabetes, heart, lung or neurological diseases, morbid obesity, pregnancy and autoimmune diseases. The main complications of swine flu were severe pneumonia and other pulmonary complications (hypoxemia, acute respiratory distress syndrome) and renal or multi-organ failure [90]. The number of deaths during the swine flu pandemic has been estimated at 0.13 to 1.87 million worldwide [74,75,76], or 0.002 to 0.03% of the global population [25].

## 6. AIDS (1981–Today)

After Congo gained independence from Belgium in 1960, the United Nations recruited French-speaking professionals from around the world to fill positions left by the colonial government and help build the new nation. Many of these professionals became infected in the Congo and brought the disease back to their countries of origin, starting the pandemic [91]. A paraffin sample taken in Congo in 1966 contained the HIV virus [92].

In June 1981, unusual cases of infectious pneumonia and skin cancer began to appear in previously healthy men in New York and California, United States. These cases were fatal, and the patients were not resistant to common infections. The cases of immune system failure and subsequent infections affected, among other social groups, homosexual men, and the syndrome was inappropriately called gay-related immune deficiency (GRID). Other social groups affected included intravenous drug users, hemophiliacs, Haitian immigrants and people who had traveled to or lived in Africa. Thus, in 1982, the syndrome was renamed Acquired Immune Deficiency Syndrome (AIDS) [13].

In 1983, two independent teams (the Pasteur Institute in Paris and the National Institutes of Health in Maryland) isolated a new retrovirus in AIDS cases, which was named the human deficiency virus (HIV) in 1986. Transmission, which at the time was still not fully understood at the time, is direct through the exchange of bodily fluids, especially blood or semen, during anal or vaginal sex, blood transfusion or sharing an intravenous injection with a needle used by more than one person, and vertically, to infants during birth or through the ingestion of breast milk. While at the beginning of the AIDS pandemic, the majority of those infected were white males, by the early 2000s, although the majority of cases were still male, the majority were black and the involvement of drug users and heterosexuals increased. As the number of infected people increased worldwide, the use of azidothymidine (AZT) was approved in 1987, which reduced the acute development of AIDS in infected individuals [13].

AIDS not only affected social practices that had no undesirable consequences (e.g., pregnancy and syphilis) since the 1960s with the introduction of the contraceptive pill and the use of antibiotics, but also emphasized safer sex practices among the general population. In addition, the stigmatization of homosexuals led to activism to empower those affected and pressure to prioritize research and care. No vaccine is commercially available, but several antiviral drugs are, making what was once a fatal disease fully manageable and with very low mortality.

To date, two types of HIV circulate, HIV-1 (circulating worldwide) and HIV-2 (circulating in West Africa). Both are the result of multiple transmissions between species of simian immunodeficiency virus (SIV) that naturally infect African primates. Most of these transmissions resulted only in viruses that did not spread to humans, but one event, involving SIV from chimpanzees in Cameroon, gave rise to today’s HIV-1 [93]. The disease likely arose from the hunting, slaughter and consumption of African primates [94]. The first AIDS case is attributed to a chimpanzee hunter in Cameroon, in 1908 [95].

## 7. Coronaviruses

Coronaviruses (CoVs) were first identified in humans in the 1960s, but have been around for approximately 300 million years, infecting bats and birds since their divergence in the Carboniferous [96]. They belong to the order Nidovirales, family Coronavididae and subfamily Orthocoronavirinae. They are enveloped viruses whose viral particle resembles a crown, hence their name. The viral envelope consists of the structural proteins M (membrane), E (envelope), and S (spicule) and the structural protein, N (nucleocapsid) is associated with the viral RNA. The viral RNA is positive single-stranded, approximately 30,000 bases in length, the second largest of all RNA viruses and the largest of the segmented viruses [97].

The large CoV genome, resulting from the acquisition of genes encoding RNA processing enzymes, the frequent recombinations, exchanges, insertions and deletions of nucleotides, and the resulting high genetic plasticity, is responsible for the possibility of CoV transmission between different species and its adaptation to new hosts. This is why all human CoVs are of animal origin and why CoV taxonomy is constantly changing [98].

There are 39 species of CoVs, divided into four genera, α- and β-coronaviruses (primarily in mammals) and γ- and δ-coronaviruses (primarily in birds) [64]. Most α- and β-CoVs are hosted by bats, and seven species also infect humans [99]. Both α- and β-CoV are cosmopolitan [100], but most scientific studies on virus circulation in bats are from China. As a result, 22 (57.9%) CoV species have been discovered in this country, where 31 (81.6%) of them are circulating [99].

In general, human coronaviruses cause respiratory disease and are mainly transmitted by air or fomites. Four of them are currently circulating endemically in the global population, causing approximately 20% of all cases of the common cold: HCoV-229E, HCoV-OC43, HCoV-NL63 and HCoV-HKU1 [101]; the other three, as soon as they emerged, caused pandemics in the last two decades: SARS-CoV and MERS-CoV and SARS-CoV-2.

### 7.1. Severe Acute Respiratory Syndrome (SARS) (2002–2004 AD)

In mid-November 2002, cases of an atypical pneumonia, thought to be caused by bacteria, began to be reported in Guangdong Province in southern China from. The cases, in turn, were caused by a novel β-CoV, the SARS-CoV. The name of the virus came from the acronym for Severe Acute Respiratory Syndrome. This virus arose from the recombination events of CoVs circulating in bats of the genus *Rhinolophus* [99], using civets (*Viverricula indica pallida*), a species of carnivore threatened by illegal trade from Southwest Asia to China, as an intermediate host for transmission between bats and humans [102]. Civets were valued as a culinary delicacy and for the production of civet coffee.

Cases pilled up, and it was not until February 2003 that an epidemic was declared, beginning the first pandemic of the 21st century. A Chinese physician treating patients in Guangdong is believed to have spread the disease to guests in a Hong Kong hotel where he was staying, who also transmitted the disease when returned to their home countries (Singapore, Vietnam, Canada and the United States) [102]. From Hong Kong, the disease quickly spread by air travel to 29 countries (Figure 18).

In addition to fever, patients experienced dry cough, headache, muscle aches, and difficulty breathing. Approximately 1/3 of the patients developed pneumonia and required mechanical ventilation, which increased the risk of death. Close contact with the patients facilitated the airborne and fomite transmission, and the short incubation period of two to seven days [103] made it easier to establish the link between the primary source and secondary infections [102].

The pandemic lasted 114 days, infected approximately 8000 people and killed approximately 800, resulting in a mortality rate of 10% [102]. Although it affected only a few people worldwide, the high mortality rate if compared to other flu diseases and the rapid spread of the disease caused great global concern and a mobilization of public health systems, particularly in Southwest Asia, which would be a valuable legacy for future pandemics.

### 7.2. Middle East Respiratory Syndrome (MERS) (2012 AD–Today)

A new beta-coronavirus was identified in July 2012, this time in Saudi Arabia, in a patient who died of acute respiratory syndrome [104]. The virus is thought to have emerged between 2007 and 2011, and it was named MERS-CoV in May 2013, after the disease it causes, Middle East Respiratory Syndrome [105]. Like SARS-CoV, it caused a pandemic as soon as it emerged, having been identified throughout the Arabian Peninsula, in addition to Saudi Arabia, and from there by air transit in the early years of the pandemic, to Algeria, Austria, China, Egypt, France, Germany, Greece, Italy, Malaysia, the Netherlands, Philippines, Republic of Korea, Thailand, Tunisia, Turkey, the United Kingdom, and the United States of America [106] (Figure 19).

After the first four years of the pandemic, MERS-CoV continues to resurface in sporadic cases [105] mainly in the Kingdom of Saudi Arabia [107]. It has infected approximately 2400 people and killed 35% of those infected [108], especially adults and males [105]. Although it is transmitted by air and fomites like the other human CoVs, human-to-human transmission is limited after zoonotic transmission. There is no evidence of sustained transmission in the community, and humans can be considered terminal hosts. However, there have been some reports of transmission outbreaks in small population clusters or hospital complexes [108].

The primary host of this virus has not been found, but it is believed to be of Chiroptera origin. The virus was identified in the feces of dromedaries (*Camelus dromedarius*) in Egypt and Saudi Arabia in 2014 [109,110], suggesting that the species is an intermediate host in the transmission between bats and humans. Occupational exposure to infected camels has been demonstrated by serologic studies, supporting the hypothesis of transmission between camels and humans. Some serologic findings of MERS-CoV in Africa, particularly in Kenya, are intriguing because although anti-MERS-CoV antibodies have been found in camels, they have not been found in people with occupational exposure to camels, but in a few with exposure to other livestock [108].

MERS is one of the world’s three ongoing pandemics, along with the seventh cholera pandemic and AIDS.

### 7.3. COVID-19 (2019–2023 AD)

In December 2019, the first suspected cases of an atypical pneumonia were reported in the city of Wuhan (China), caused by a new coronavirus, later named SARS-CoV-2 (a new β-CoV). Since then, the world has faced the largest pandemic in a hundred years, caused by a disease then called COVID-19. Virtually all countries and territories were affected within 6 months, with the exception of some oceanic islands (Figure 20).

The global spread of SARS-CoV-2 was extremely rapid, due to air travel. Once a positive case was identified, the virus spread rapidly throughout the community, unchecked by sanitary barriers. Successive failures in public health decision-making by both the WHO and individual countries, combined with the rapid spread of the virus, have plunged the world into the worst public health crisis of the 21st century. Never before in human history has a pandemic spread to so many areas in such a short space of time. The countries with the highest cumulative numbers of cases and deaths were the United States, India and Brazil. While the epicenter of the pandemic was initially in Europe, it quickly shifted to the Americas. The low number of cases and deaths on the African continent was probably due to a lack of structured health services and low levels of testing. In East Asia, on the other hand, the low number of positive cases was the result of an efficient pandemic preparedness structure [111]. Precisely in the area of the planet that had suffered the effects of SARS and MERS. By March 2024, 775 million cases and approximately 7 million deaths had been reported worldwide [112]. SARS-CoV-2 is neither the most contagious nor the deadliest of the other pandemic pathogens, but it has caused a high death toll due to several factors, including its mode of transmission, intense air travel, and the inability to contain it. The pandemic not only impacted the global economy, but has also overwhelmed public health systems, with economic and geopolitical repercussions long after its end.

In January 2020, a group of Chinese researchers led by one of the world’s leading experts in the study of CoVs in bats, Dr. Zheng-Li Shi, of the Center for Emerging Infectious Diseases at the Wuhan Institute of Virology, hypothesized the origin of SARS-CoV-2 in a chiropteran reservoir of the family Rhinolophidae, in line with the natural history of other HCoVs. The genetic sequence of a sample, designated RaTG13, obtained from a rectal swab collected from a horseshoe bat (*Rhinolophus affinis*) in 2013, in Yunnan Province (southern China) had 96.2% similarity to the genetic sequences of SARS-CoV-2 obtained in December 2019 in Wuhan [113].

Three hypotheses have been proposed for the origin of SARS-CoV-2: (i) natural selection in animal hosts before zoonotic transfer; (ii) natural selection in humans after zoonotic transfer; (iii) selection during passage in cell culture or animal models in the laboratory (viral engineering). The second hypothesis was refuted by the inability of SARS-CoV-like CoVs circulating in bats to bind to human cell receptors, i.e., the transmission of SARS-CoV-2 directly from the chiropteran reservoir to humans was unlikely. It would be necessary to acquire different CoV characteristics by recombination in another (intermediate) host in order to optimize for binding to human cell receptors. The third hypothesis was also rejected because the acquisition of mutations necessary for binding to human cells could only occur in the presence of an immune system, i.e., by natural selection, and not in cell culture [114].

The pangolin (*Manis javanica*), one of the most endangered mammals, was the likely intermediate host for the transmission of the SARS-CoV-2 precursor from bats to humans. This hypothesis was formulated on the basis of two CoV samples obtained from pangolins intercepted during anti-trafficking operations: one collected in 2019 in Guangdong, China, with 91.2% similarity to SARS-CoV-2 samples isolated from humans, and another, in 2020 in Guangxi, China, with 85.4% similarity. Although less similar than the RaTG13 sample from the horseshoe bat, these samples are virtually 100% similar in the region covering the S protein binding site of the virus to the human ACE-2 receptor [115]. Regardless of the complete elucidation of the natural history of SARS-CoV-2, it is now a human virus because new zoonotic jumps, if they existed, would likely have been insignificant in changing the state of the pandemic. Infected humans could have been sources of infection for other species, meaning that COVID-19 could have been an anthropozoonosis [116,117].

Although widely reported in the media as the source of SARS-CoV-2, the Huanan live animal market in Wuhan, which was in early January 2020, did not sell live or dead bats, but it did sell pangolins, according to Chinese government reports. In contrast, the first cases of COVID-19, detected in December 2019, had no epidemiologic history of contact with the market [118,119]. Phylogenetic analyses have suggested that the source of SARS-CoV-2 was outside the market. According to these analyses, smaller population groups were infected earlier, and these brought the virus to the market [120].

This information raised the hypothesis that hunters, smugglers or recipients of endangered animals destined for Chinese markets may have become infected with SARS-CoV-2, or even that infected animals delivered to China may have infected middlemen and these people may then have been the founding hosts of the pandemic. By visiting the live animal market in Huanan, either as consumers or suppliers, these populations may have infected more people there, spreading the virus to the city of Wuhan and from there to the rest of the world. Wuhan is a populous city and well connected to global centers. Infected people, whether symptomatic or not, could quickly have traveled to all continents, spreading SARS-CoV-2 around the world [111]. Phylogenetic analyses of complete SARS-CoV-2 genomes isolated from human patients support this spread theory [121].

Declared a pandemic by the WHO in March 2020, COVID-19 lasted just over three years and was declared over in May 2023. However, despite vaccination campaigns, cases and deaths from COVID-19 are still being reported around the world, leading to the belief that the disease could become endemic, like the other CoVs.

## 8. Discussion

A pandemic could be defined as an emerging or re-emerging disease with global spread, i.e., an outbreak that, after affecting a specific region, spreads locally and then to different continents with sustained human-to-human transmission, resulting in increased morbidity and/or mortality and social, economic and/or political disruption. This definition could guide preparations for the next pandemics.

Unfortunately, even with the knowledge accumulated from past pandemics, it is impossible to predict when and where the next pandemic will begin. In the future, however, humanity will experience concurrent, rapidly spreading pandemics that, while less lethal, will cause social, economic, and political disruption. The pathogens involved will most likely be zoonotic, emerging RNA viruses with low pathogenicity but high transmissibility.

It is important to depict pandemics by maintaining the time scale and magnitude of each pandemic for a better comparison. The Spanish flu was undoubtedly the pandemic that killed the most people in the shortest amount of time, although it was not the deadliest pandemic and is therefore considered by some authors to be the most important pandemic in human history [122]. The deadliest pandemic was the Black Death, which killed twice as many people as the Spanish flu, but in a period 7.5-fold longer (Appendix A).

It is also possible to see the concentration of pandemics after the beginning of the 19th century. Although they were less deadly than earlier pandemics (with the exception of the Spanish flu), their frequency was significantly higher, with up to three pandemics occurring simultaneously on several occasions. Since the beginning of the 19th century, humanity has experienced at least one pandemic during this entire period. Although the COVID-19 pandemic has ended in 2023, three pandemics were still occurred: the 7th cholera pandemic, AIDS, and MERS.

Moreover, the pandemic timeline reveals changes in temporal, epidemiological, economic and social patterns. For example, it is possible to analyze the time between pandemics in years or in generation time (Appendix A). Surprisingly, the human generation time of 26.9 years has remained virtually constant over the past 250,000 years [123]. Until the start of the first cholera pandemic (1817 AD), the time between pandemics was quite long, ranging from 182 to 797 years, or 6.8 to 29.6 generations. However, after the first cholera pandemic, the time between pandemics decreased significantly, ranging from 3 to 39 years, or 0.1 to 1.4 generations. In general, the time between pandemics during this period was less than one generation, and only between the Spanish flu (1918) and the Asian flu (1957) it was longer (1.4 generations) than one generation.

The first five cholera pandemics were separated by only a few years, and it is difficult to distinguish between them. They started in the same region and the intervals between them were extremely short, ranging from two to six years. Considering the first six cholera pandemics as a single pandemic, the interpandemic period from the first cholera pandemic (1817 AD) the Spanish flu pandemic (1918 AD) was approximately one generation (range 1.08 to 1.4). After the early of the 19th century, each generation experienced two to three different pandemics, but after the 1950s, the same generation sometimes experienced up to four different pandemics.

The speed at which people move around the world has increased exponentially since the 19th century. While the average daily distance for all modes of transportation was approximately 6 km/capita/day, it increased to over 100 km/capita/day in the United States in the early 21st century. This was due to the decline in the use of horses by 1920 and their replacement by automobiles, buses, and motorcycles. The use of air travel began in the late 1930s, accompanied by a decline in the use of trains [124]. These changes significantly altered the rates of human-to-human contact, including infectious contact.

Another striking aspect of the pandemic timeline is the variation in the number of deaths. Until the first cholera pandemic, the proportion of the world’s population killed in each pandemic was greater than or equal to 5%, reaching 51% during the Black Death. The Justinian Plague and the Black Death were bottleneck events, two moments when we came closest to extinction. After the first cholera pandemic, however, the proportion was at most 1%, with the exception of the Spanish flu, which killed 5.4% of the world’s population (Appendix A). Disturbances in the vegetative growth of the human population caused by pandemics were more significant until the early 20th century, when pandemics no longer disrupted the logarithmic pattern of population growth (Appendix A).

The pattern of transmission of pandemic pathogens has also changed. Until the early 20th century, most pandemics were caused by bacteria (Appendix A) transmitted through the fecal–oral route, due to a lack of hygiene and minimally adequate sanitation. Moreover, until the end of the 18th century, medicine was basically the same as it had been since ancient times. It was not until the beginning of the 19th century that the incorporation of scientific knowledge led to improved effectiveness in both diagnosis and treatment [125]. Although the first antibiotic was introduced in 1910, they were not popularized until the 1950s [126]. Furthermore, the use of vaccines did not begin until the end of the 17th century, with the invention of the smallpox vaccine [20]. It was not until almost a century later that other vaccines began to be used: the plague vaccine in 1892 [51], the cholera vaccine in 1896 [57], the influenza vaccine in 1942 [87], and the coronavirus vaccines since 2005 [127].

Viruses have become responsible for the most recent pandemics. Among viruses, those with RNA have greater adaptive capacity to different environments and selective pressures. This characteristic stems from high mutation and substitution rates compared to DNA viruses [128]. Among the viruses, those with RNA are potentially the most involved in zoonoses. In addition to their high diversity and adaptive capacity, anthropogenic changes that increase not only population growth, but also interspecific contact rates and, consequently, the exchange of pathogens, increase the risk of RNA viruses becoming pandemic [128]. Recent pandemics have been caused by emerging airborne viruses that have used the air travel to spread. Therefore, their surveillance should be a local and global priority.

There is much speculation that the next epidemic will be caused by an RNA virus, but insidiously and quietly, it may have another cause. Antimicrobial resistance is neglected and obscure and its burden is not properly understood. Healthcare spending should be primarily directed at preventing infections, and when treatment is necessary, antibiotics should be used responsibly and in a controlled manner [129]. Antimicrobial resistance must be taken seriously by local, national and global health authorities.

Pandemic pathogens are not among the most transmissible human pathogens, i.e., they do not have the highest baseline reproducibility rates (*R*_0_) (Appendix A). Although they have different case fatality rates, this is not the only factor in their emergence. The reason why they became pandemics may be related to the perfect timing of human socioeconomic and health conditions and, consequently, the ease of fecal–oral transmission and the lack of minimal sanitary conditions. This is the case with smallpox, cholera and plague. Cholera is one of the most rapidly fatal human diseases, with a mortality rate of over 50% if left untreated [54]. It has an *R*_0_ intermediate between smallpox and plague, yet it has caused seven pandemics, one of which is still ongoing at the time of writing this work. Conversely, both bubonic and pneumonic plague cause 60% mortality without treatment [32], but have low *R*_0_.

Another reason for the pandemic behavior may be related to the proportion of asymptomatic individuals and the ease of airborne transmission. For example, although SARS, MERS and COVID had similar incubation periods and *R*_0_*s*, they differed in their mortality rates. The SARS and MERS pandemics were not very large, although MERS is still occurring at the time of writing. COVID-19, also caused by a CoV, spread more widely than the other coronavirus epidemics and caused more deaths, although it had a low mortality rate but a higher *R*_0_. The incubation period for COVID-19 was much longer, and it was difficult to detect those infected early, which allowed it to spread so quickly [130]. Compared to coronaviruses, the Spanish flu, had a shorter incubation period and a higher proportion of asymptomatic individuals, despite a similar *R*_0_. As a result, it killed more people despite a lower mortality rate [130]. The lack of individual protection and isolation measures and the absence of antibiotics must be taken into account, as most deaths from Spanish flu were caused by secondary infections [63]. Smallpox has one of the highest *R*_0_*s* among human pathogens and a long incubation period, despite a mortality rate of approximately 30%. It is the only disease that has prompted global eradication efforts to date.

Of the 1415 known human pathogens, 271 are viruses or prions, 538 are bacteria or rickettsiae, 307 are fungi, 287 are helminths and 66 are protozoa. The majority of these pathogens (868 or 61.3%) are of animal origin, i.e., zoonotic [131]. The proximity to domestic animals, mainly mammals, since 10,000 years ago, either by consuming them, using their by-products or raising them as pets, is the probable cause of the exchange of pathogens between them.

Between 1975 and 2000, 175 emerging pathogens were identified, of which 132 (75.4%) were zoonotic [131]. Four of these became pandemic: HIV, SARS-CoV, MERS-CoV and SARS-CoV-2 (Appendix A).

Of the 616 known pathogens of domestic ruminants and the 374 pathogens of domestic carnivores (dogs and cats), 77.3% and 90%, respectively, affect multiple host species, including humans [132]. However, this coevolution of humans with domesticated species has attenuated the severity of the pathogens. Contact with wild animals, although rarer, can result in disproportionately severe diseases when zoonotic pathogens are transmitted. Anthropogenic changes, especially urbanization, changes in land use and exploitation of natural environments have increased the likelihood of this type of contact.

Of the 95 human viruses of zoonotic origin, 91% are of wild origin and only 34% are of domestic origin [133]. There is also a positive correlation between the species richness of mammalian orders and the richness of zoonotic pathogens [134]. In descending order, Rodentia, Chiroptera, Carnivora and Primates are the mammalian orders with the highest number of hosts of zoonotic pathogens. Because Rodentia and Chiroptera have approximately half the species richness of mammals, they have a wide geographic distribution and therefore a wide host plasticity for pathogens.

Transmission of zoonotic pathogens from mammals occurs directly or indirectly, often around human settlements in agricultural areas or through occupational exposure. Hunting and/or consumption of wild animals sold in markets, which is very common in Latin America, Africa and Asia, significantly increases the transmission of pathogens between animals and humans and facilitates subsequent human-to-human transmission [133]. This assumption is based on the greater likelihood that a pathogen will emerge when both hosts and pathogens are locally dispersed and that it will spread independently of the host, e.g., through the air [5].

While the greatest abundance of potential mammalian hosts for zoonotic pathogens is observed in megadiverse locations, in Central and South America, Central and East Africa, Eastern Europe and part of Western Europe, in addition to continental Southwest Asia, reports of health emergencies were not always positively correlated with this abundance, and in some cases were negatively correlated. Nevertheless, reports of emerging diseases have been reported in populations living in areas of high biodiversity [134]. Failure to establish correlations may be due to inadequate surveillance and monitoring efforts for pathogens in wildlife [135], making it difficult to predict when and where the next emerging disease will occur or the next pandemic will occur. There is a clear need for multidisciplinary integration to properly assess, mitigate and communicate the risks of zoonotic pathogen transmission.

Emerging and re-emerging diseases are often associated with poverty, vulnerability, hunger and low levels of health care, conditions that are more prevalent in poor countries [136]. A pandemic has the potential to generate intense scientific and political debate, especially in the media and social networks, as seen during the COVID-19 pandemic. This situation requires surveillance systems to have a clear vision of the epidemiology of the pathogen involved, with the aim of stopping the pandemic before it becomes uncontrollable. The role of institutions such as the WHO is essential in providing guidance to countries to stop the next pandemic using the same criteria, for example, clear case definition, genomic surveillance, traceability [136] and diagnostic criteria.

## 9. Conclusions

A pandemic can be caused by an emerging or re-emerging disease with global spread, i.e., an outbreak that, after affecting a specific region, spreads locally and then to different continents with sustained human-to-human transmission, resulting in increased morbidity and/or mortality and social, economic and/or political disruption. This definition could guide preparations for the next pandemics.

Even with the accumulated data from previous pandemics, it is virtually impossible to predict where and when the next pandemic will occur. However, it is likely that humanity will experience simultaneous, rapidly spreading pandemics that, while less lethal, will cause social, economic, and political disruption. The pathogens involved will most likely be zoonotic, emerging RNA viruses with low pathogenicity but high transmissibility. For these reasons, preventing future pandemics requires a multifaceted approach that addresses several interrelated factors. Strengthening zoonotic disease surveillance systems is essential to detect and respond to emerging threats. In addition, systematic monitoring of pathogens in wildlife can provide critical insights into potential disease reservoirs. Efforts to conserve biodiversity are essential not only for ecological balance, but also to reduce the risk of zoonotic spillover. Enforcement of laws against environmental crimes, particularly wildlife trafficking, is essential to curb the illegal trade that facilitates disease transmission. In addition, targeted measures, such as implementing surveillance and risk reduction strategies for high-risk populations and providing alternative protein sources, can mitigate human–wildlife interactions. Closing live animal markets in regions of high biodiversity and trade activity is essential to minimize opportunities for disease transmission. Finally, improving surveillance systems for zoonotic pathogens in wildlife, coupled with the establishment of biorepositories, can improve our understanding of disease dynamics and aid in proactive prevention measures.

By addressing these key issues comprehensively and collaboratively on a global scale, we can significantly reduce the risk of future pandemics and protect both human and environmental health. It is imperative that policy makers, researchers and stakeholders work together to implement these strategies effectively and sustainably.

## Figures and Tables

**Figure 1 microorganisms-12-01802-f001:**
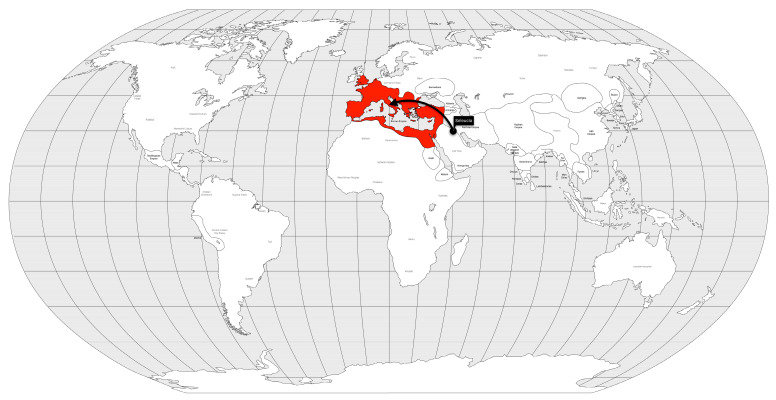
Dissemination of the Antonine Plague from 165 to 180 AD.

**Figure 2 microorganisms-12-01802-f002:**
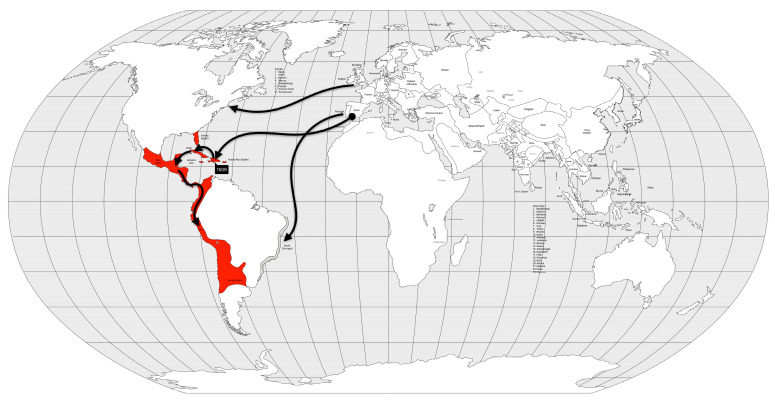
Dissemination of the smallpox from 1520 to 1880 AD.

**Figure 3 microorganisms-12-01802-f003:**
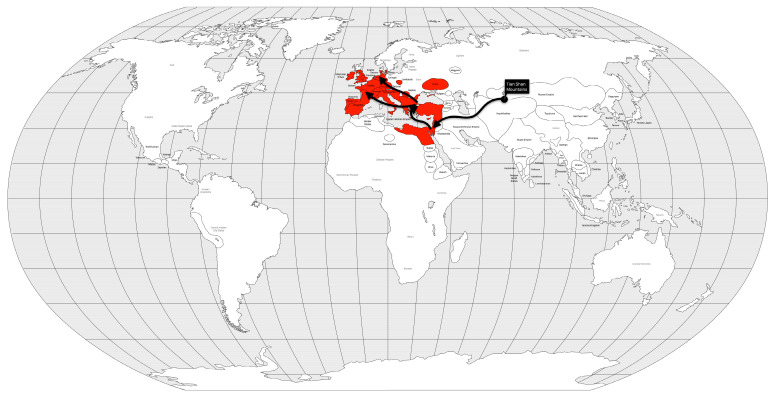
Dissemination of the Justinian Plague from 541 to 544 AD.

**Figure 4 microorganisms-12-01802-f004:**
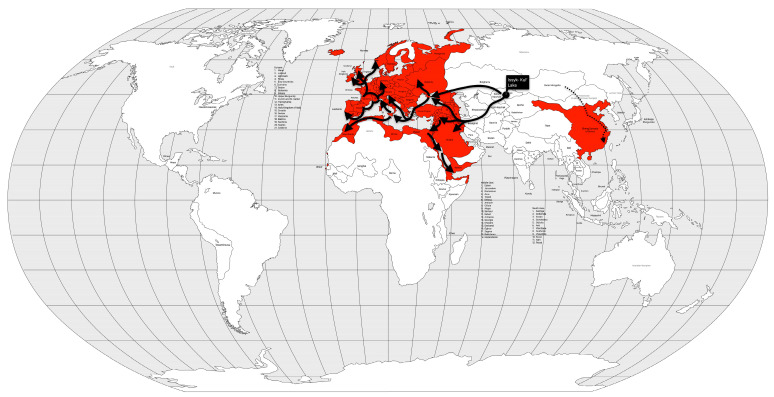
Dissemination of the Black Death from 1338 to 1353 AD.

**Figure 5 microorganisms-12-01802-f005:**
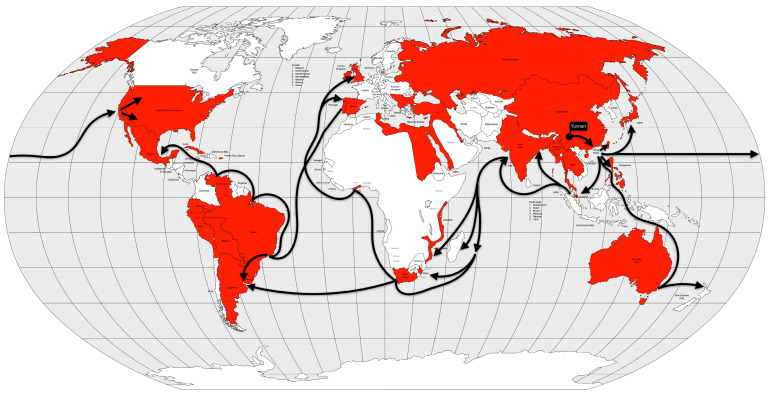
Dissemination of the third plague pandemic from 1855 to 1960 AD.

**Figure 6 microorganisms-12-01802-f006:**
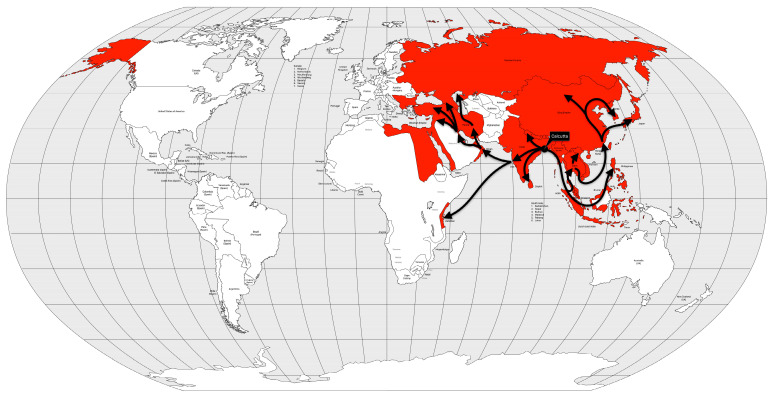
Dissemination of the first cholera pandemic from 1817 to 1824 AD.

**Figure 7 microorganisms-12-01802-f007:**
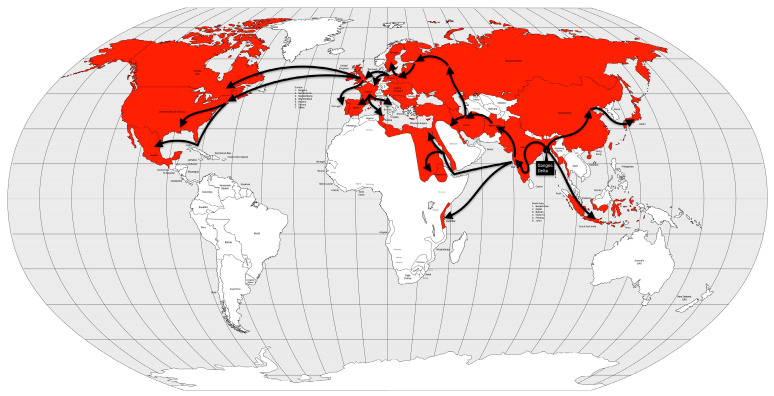
Dissemination of the second cholera pandemic from 1826 to 1835 AD.

**Figure 8 microorganisms-12-01802-f008:**
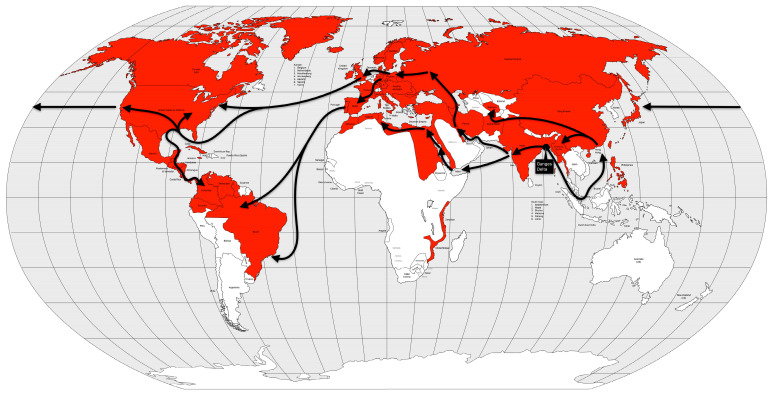
Dissemination of the third cholera pandemic from 1839 to 1860 AD.

**Figure 9 microorganisms-12-01802-f009:**
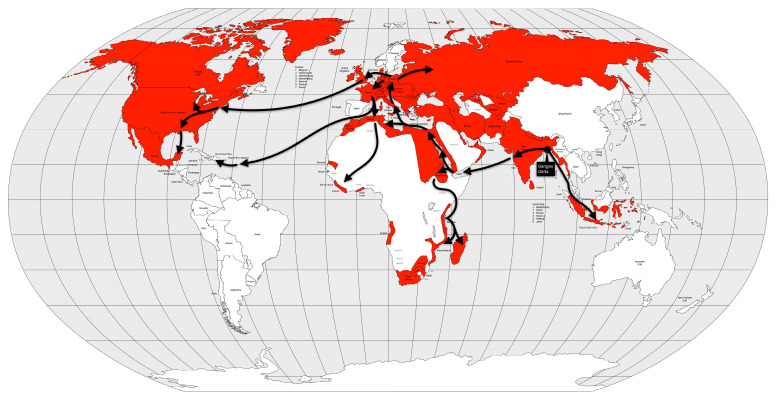
Dissemination of the fourth cholera pandemic from 1863 to 1875 AD.

**Figure 10 microorganisms-12-01802-f010:**
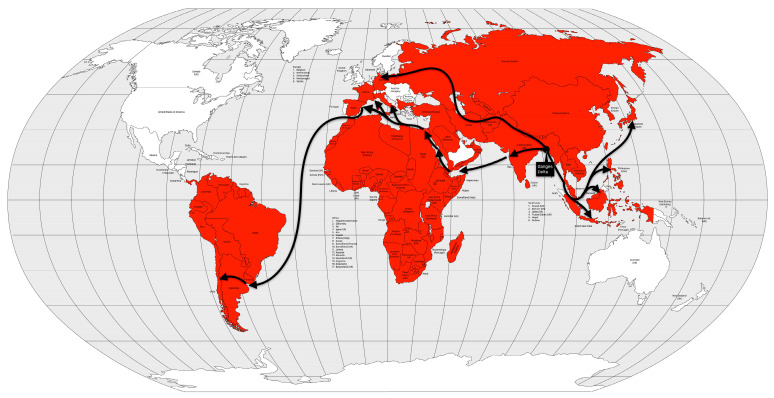
Dissemination of the fifth cholera pandemic from 1881 to 1896 AD.

**Figure 11 microorganisms-12-01802-f011:**
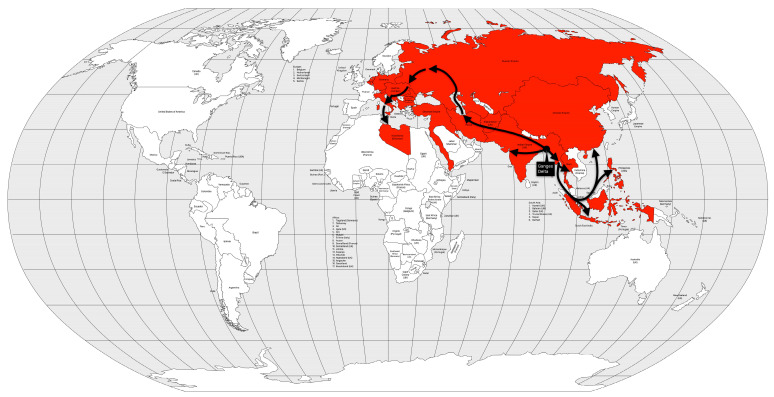
Dissemination of the sixth cholera pandemic from 1899 to 1923 AD.

**Figure 12 microorganisms-12-01802-f012:**
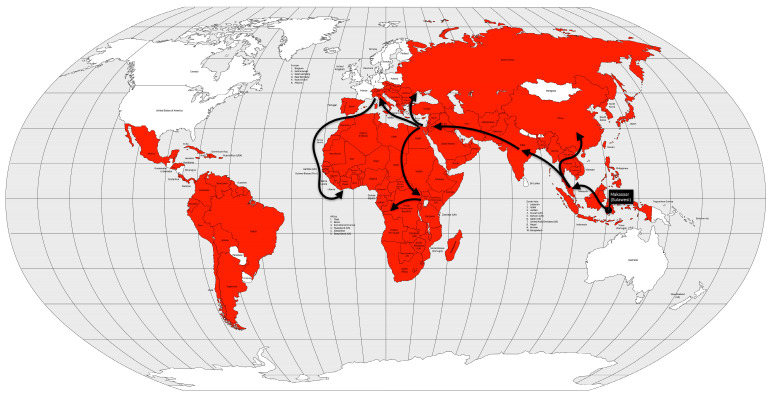
Dissemination of the seventh cholera pandemic from 1961 AD to today.

**Figure 13 microorganisms-12-01802-f013:**
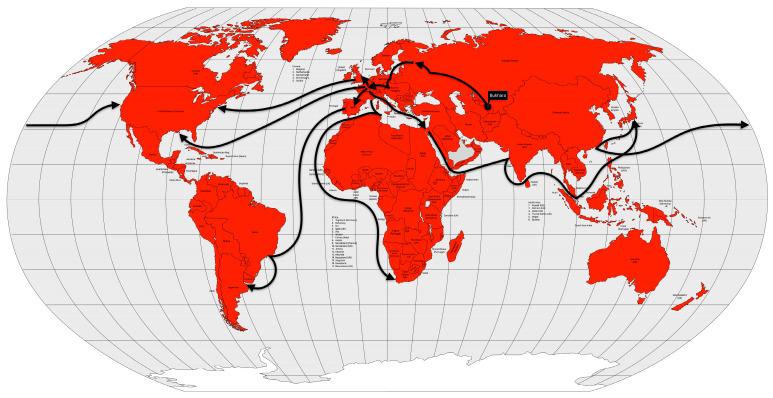
Dissemination of the Russian flu pandemic from 1889 to 1890 AD.

**Figure 14 microorganisms-12-01802-f014:**
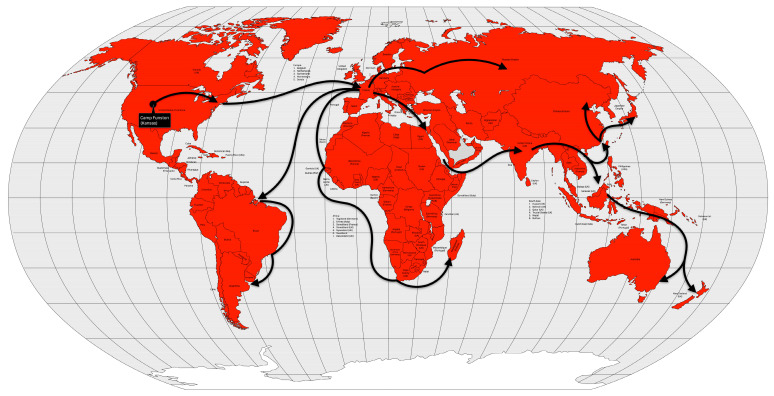
Dissemination of the Spanish flu pandemic from 1918 to 1919 AD.

**Figure 15 microorganisms-12-01802-f015:**
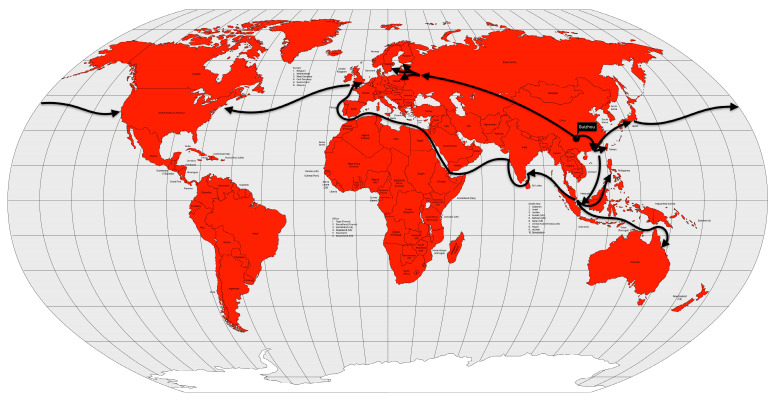
Dissemination of the Asian flu pandemic from 1957 to 1958 AD.

**Figure 16 microorganisms-12-01802-f016:**
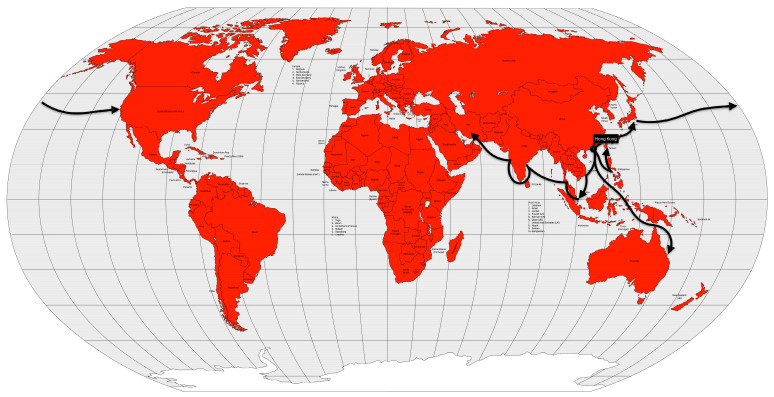
Dissemination of the Hong Kong flu pandemic from 1968 to 1969 AD.

**Figure 17 microorganisms-12-01802-f017:**
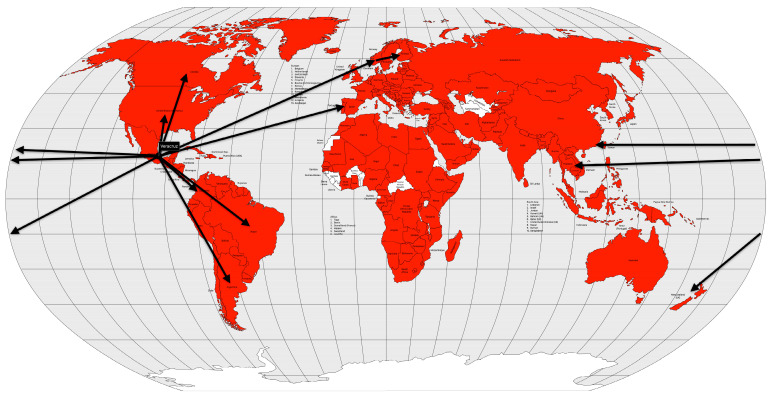
Dissemination of the Swine flu pandemic from 2009 to 2010 AD.

**Figure 18 microorganisms-12-01802-f018:**
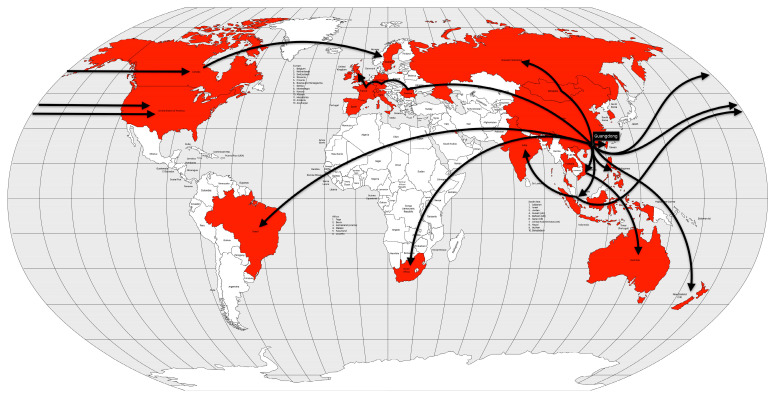
Dissemination of the SARS pandemic from 2002 to 2004 AD.

**Figure 19 microorganisms-12-01802-f019:**
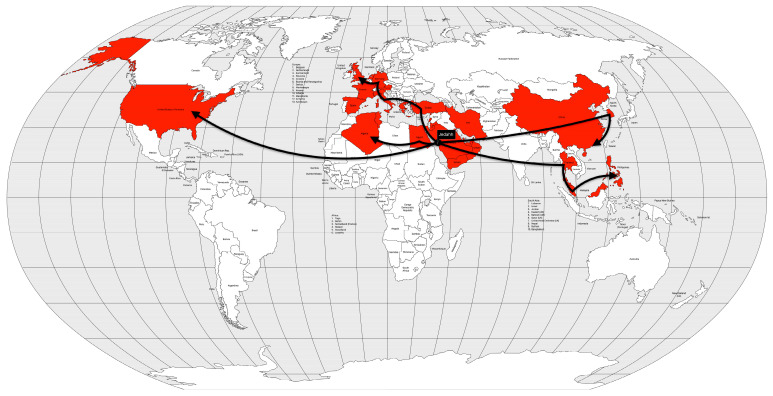
Dissemination of the MERS pandemic from 2012 AD to today.

**Figure 20 microorganisms-12-01802-f020:**
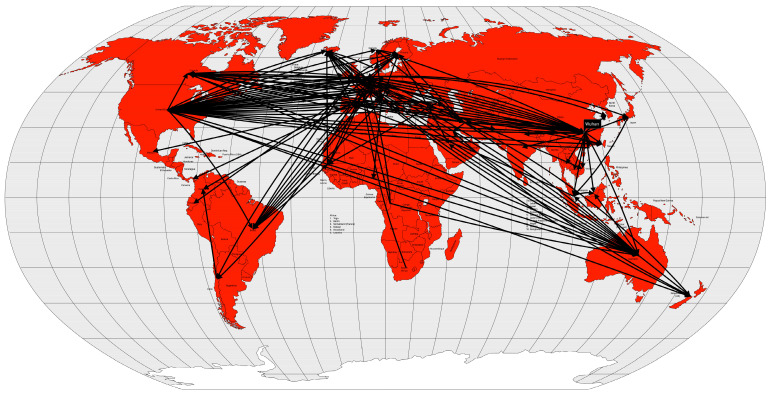
Dissemination of the COVID-19 pandemic from 2019 to 2023 AD.

## Data Availability

No new data were created or analyzed in this study. Data sharing is not applicable to this article.

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
