# Peer review of "Towards a Comprehensive Definition of Pandemics and Strategies for Prevention: A Historical Review and Future Perspectives"

_microorganisms, 2024, doi:10.3390/microorganisms12091802_

Round 1

Reviewer 1 Report

Comments and Suggestions for Authors

Introduction

Comment: Introduction could also add the WHO definition as well

Line 36. ………….even the metapopulation structure.

Comment: provide the reference

Lines 58-60. Provide the reference

Lines 68-72. Provide the reference

Line 73. By this definition…. Which definition you are referring to?

Lines 77-81. Provide the reference

Line 98. The sentence is incomplete. Please revise it?

Comments on MERS-CoV. It seems that you have disregarded several publications published in Gulf Cooperation Council (GCC) states that provide detailed explanations about the spread of the ailment, which originated from Saudi Arabia. Please evaluate these literary works.

Results

Comment: The maps provide an excellent illustration, but it will be fascinating to use highlite in RED to pinpoint the disease's potential cause.

Discussions

Avoid presenting data and figures ie Figures 21, 23 and 24

Focus as much as possible discussing the results and the way forward

The conclusions do not capture the essence of the study titles, such as the review of the pandemic definition. The present conclusions capture any novel concepts. Please revisit

Author Response

Response letter of the manuscript entitled:

“Towards a comprehensive definition of pandemics and strategies for prevention: a historical review and future perspectives”

Reviewer #1:

Introduction:

Comment: Introduction could also add the WHO definition as well

Response: I had an old WHO reference, which is not available anymore, after COVID-19. Anyway, there is no WHO definition for pandemic since.

Line 36. ………….even the metapopulation structure. Comment: provide the reference

Response: A reference was added.

Lines 58-60. Provide the reference

Response: There is no reference, since this is an assumption. The previous paragraph has been rewritten, changing must and can for should.

Lines 68-72. Provide the reference

Response: There is no reference, since this entire paragraph is a tentative definition of pandemic.

Line 73. By this definition…. Which definition you are referring to?

Response: I meant “Based on the above (definitions)”. Rewritten. I also rewrote the entire paragraph to make it clear that this  is actually the (tentative) proposal of a new and broader definition of pandemic.

Lines 77-81. Provide the reference

Response: There is no reference to include, since the 22 pandemics described in this manuscript are based in the “tentative definition” of pandemic. Even so, it was rewritten to bring this idea clearer.

Line 98. The sentence is incomplete. Please revise it?

Response: It was incomplete indeed. Rewritten.

Comments on MERS-CoV. It seems that you have disregarded several publications published in Gulf Cooperation Council (GCC) states that provide detailed explanations about the spread of the ailment, which originated from Saudi Arabia. Please evaluate these literary works.

Response: A reference of compiled information about MERS in the Middle East was included and some parts of this section were updated. Thank you for the suggestion.

Results:

Comment: The maps provide an excellent illustration, but it will be fascinating to use highlight in RED to pinpoint the disease's potential cause.

Response: Suggestion accepted. All maps were changed.

Discussion:

Avoid presenting data and figures ie Figures 21, 23 and 24

Response: Thank you, the figures 21-26 were placed in the Supplementary Material.

Focus as much as possible discussing the results and the way forward

Response: I have included two paragraphs at the beginning and some changes throughout the Discussion to improve this impression. Thank you.

Conclusion:

The conclusions do not capture the essence of the study titles, such as the review of the pandemic definition.

Response: The conclusion was reformulated, in light of the reviewer’s comments.

The present conclusions capture any novel concepts. Please revisit

Response: The conclusion was reformulated, in light of the reviewer’s comments.

Thank you very much for your review.

The manuscript was greatly improved with your comments.

Reviewer 2 Report

Comments and Suggestions for Authors

See attached file

Comments on the Quality of English Language

Minor spell check required

Author Response

Response letter of the manuscript entitled:

“Towards a comprehensive definition of pandemics and strategies for prevention: a historical review and future perspectives”

Reviewer #2:

General comments:

The abstract completely lacks the chapter structure but well summarizes the content of the review.

Response: Thank you. Since there is no requirements for headings in the abstract, the abstract was maintained as is.

In the discussion I suggest developing a part for prevention and public health preparedness, for this I suggest this article: Overview of case definitions and contact tracing indications in the 2022 monkeypox outbreak. Guarducci G, Porchia BR, Lorenzini C, Nante N. - Infez Med. 2023 Mar 1;31(1):13-19. doi: 10.53854/liim-3101-3.

Response: This reference was very enlightening and important for improving the discussion of the manuscript. Thank you very much for your suggestion.

The references are numerous, but they should be reported in the manner required by the journal, indicating in the text the number [n] of the corresponding citation in the list of references at the end of the manuscript.

Response: The references were reformulated according to the journal’s requirements.

Line 45: “sustained transmission” or “emergence of new pathogens” (in quotes).

Response: Rewritten.

Line 1243: the pandemic line corresponds to figure 22, not 21.

Response: Thank you. Figures in the Discussion section were placed in the Supplementary Material.

Graphs and tables are easy to read and provide useful support in understanding the phenomenon described, but I recommend you put some of them (at least the ones in the discussion) as supplementary materials.

Response: Figures 21-26 (previously in the Discussion section, were placed in the Supplementary Material.

Conclusions:

Conclusions are concise and incisive. Consistent with the evidence and arguments presented.

Response: Thank you.

Hints for future monitoring and prevention strategies are provided.

Response: Thank you.

The bibliography is very extensive and well done, containing many recent articles and other older ones.

Response: Thank you.

It could be implemented with the article I suggested you before.

Response: The reference was included as suggested.

Thank you very much for your review.

The manuscript was greatly improved with your comments.

Round 2

Reviewer 1 Report

Comments and Suggestions for Authors

NIL

Reviewer 2 Report

Comments and Suggestions for Authors

Thanks to the author for following my suggestions. The manuscript is still a bit too long, but better than before.